# Redox-homogeneous, gel electrolyte-embedded high-mass-loading cathodes for high-energy lithium metal batteries

Jung-Hui Kim [1,5], Ju-Myung Kim[1,3,5], Seok-Kyu Cho[1,4], Nag-Young Kim[2] & Sang-Young Lee [2✉]

Lithium metal batteries have higher theoretical energy than their Li-ion counterparts, where graphite is used at the anode. However, one of the main stumbling blocks in developing practical Li metal batteries is the lack of cathodes with high-mass-loading capable of delivering highly reversible redox reactions. To overcome this issue, here we report an electrode structure that incorporates a UV-cured non-aqueous gel electrolyte and a cathode where the $LiNi_{0.8}Co_{0.1}Mn_{0.1}O_2$ active material is contained in an electron-conductive matrix produced via simultaneous electrospinning and electrospraying. This peculiar structure prevents the solvent-drying-triggered non-uniform distribution of electrode components and shortens the time for cell aging while improving the overall redox homogeneity. Moreover, the electron-conductive matrix eliminates the use of the metal current collector. When a cathode with a mass loading of 60 mg cm$^{-2}$ is coupled with a 100 μm thick Li metal electrode using additional non-aqueous fluorinated electrolyte solution in lab-scale pouch cell configuration, a specific energy and energy density of 321 Wh kg$^{-1}$ and 772 Wh L$^{-1}$ (based on the total mass of the cell), respectively, can be delivered in the initial cycle at 0.1 C (i.e., 1.2 mA cm$^{-2}$) and 25 °C.

[1] Department of Energy Engineering, School of Energy and Chemical Engineering, Ulsan National Institute of Science and Technology (UNIST), Ulsan 44919, Republic of Korea. [2] Department of Chemical and Biomolecular Engineering, Yonsei University, Seoul 03722, Republic of Korea. [3] Present address: Energy and Environment Directorate, Pacific Northwest National Laboratory, Richland, WA 99353, USA. [4] Present address: Battery Materials Research Center, Research Institute of Industrial Science and Technology (RIST), Pohang 37673, Republic of Korea. [5] These authors contributed equally: Jung-Hui Kim, Ju-Myung Kim. ✉email: syleek@yonsei.ac.kr

The promise of forthcoming smart portable electronic devices, electronic vehicles (EVs), drones, and the Internet of things has spurred the relentless pursuit of high-energy-density lithium (Li) batteries[1–3]. Many previous studies implemented to achieve this goal were devoted to the synthesis and modification of electrode active materials and electrolytes[4–6]. Along with these material-based works, the design of high-mass-loading electrodes has recently garnered considerable attention as a facile and scalable architectural strategy[7,8].

A formidable challenge facing high-mass-loading electrodes is the acquisition of uniform ion/electron conduction pathways throughout their through-thickness direction without structural disruption[8–12]. One notable result was the anisotropic-structured porous electrodes that provided short tortuous migration paths of the electrolytes. These electrodes were fabricated by a subtractive method based on aligned sacrificial templates (magnetized nylon rods/emulsion droplets[13] and NaCl salts[14]) or an additive method (three-dimensional (3D) printing[15–17], ice templating[18,19], and wood templating[20,21]). Unfortunately, the subtractive and additive methods suffered from an inevitable increase in electrode porosity. Moreover, larger amounts of electrolyte are needed to fill the electrode pores[8]. Consequently, unwanted loss of energy density and specific energy was encountered in the resulting cells. As a different approach, electron-conducting frameworks were introduced in the electrodes. Infiltrating electrode slurries into 3D conductive scaffolds[22–24] or integrating electrode materials with conductive percolation networks[25–28] improved the electronic conductivity and mechanical robustness of the electrodes. However, the weights and volumes of the electron-conducting frameworks themselves were excessively large, thus decreasing the mass loadings of electrode active materials in the electrodes[29]. Additionally, the complicated and high-cost manufacturing processes hindered the practical and scalable production of high-mass-loading electrodes.

Most of the research works described above tended to focus only on the absolute values of mass loadings of electrodes, with little attention to the ultimate goal of achieving cells with high energy content. As an indispensable prerequisite to resolve this challenging issue, the theoretical specific capacities of electrode active materials should be fully utilized throughout the entire region of electrodes, together with minimal usage of other cell components (such as electrolytes and metallic-foil current collectors).

Here, we demonstrate an electrode production strategy based on a bicontinuous electron/ion conduction network-embedded quasi-solid-state (BNQS) architecture. Coupling ultraviolet (UV)-cured gel electrolytes with single-walled carbon nanotube (SWCNT)-based electroconductive-mat interlayers enables the BNQS electrodes to exhibit homogeneous redox reactions in the through-thickness direction. As a proof of concept, we choose high-capacity Ni-rich layered transition metal oxide ($LiNi_{0.8}Co_{0.1}Mn_{0.1}O_2$, NCM811) particles as a model electrode active material[30,31]. The UV-curable gel electrolyte precursors are mixed with NCM811 particles and conductive additives without using any typical processing solvents, such as N-methyl pyrrolidone (NMP), thus enabling the removal of solvent-drying steps in the electrode manufacturing and a shorter time for cell aging. Drying of processing solvents during electrode fabrication triggers an uneven and random distribution of electrode components[32]. Therefore, the BNQS electrode is expected to allow a uniform distribution of NCM811 particles, carbon conductive additives, and gel electrolytes. Meanwhile, the electroconductive-mat interlayers are placed along the through-thickness direction of the BNQS electrodes, thereby facilitating cross-sectional electron transport and eliminating metallic-foil current collectors.

The BNQS cathode exhibits a high areal-mass (solely including NCM811) loading ($60 \, mg \, cm^{-2}$ (thickness ~ 315 μm) corresponding to an areal capacity of $12.3 \, mAh \, cm^{-2}$ at 4.4 V). Furthermore, the BNQS cathode provides improvements in the redox reaction kinetics and cyclability at high areal-mass-loading levels. Notably, the theoretical specific capacity ($205 \, mAh \, g^{-1}$)[30,31] of NCM811 is almost realized in the BNQS cathodes over a wide range of mass loadings, demonstrating the viable role of the ion/electron conduction networks for the full utilization of the electrochemical activity of NCM811. We also assembled and tested a pouch cell by coupling a BNQS positive electrode ($12.1 \, mAh \, cm^{-2}$) and 100 μm thick Li metal negative electrode capable of delivering a specific energy and energy density of $321 \, Wh \, kg^{-1}$ and $772 \, Wh \, L^{-1}$ (based on the total mass of the cell) in the initial cell cycling at 0.1 C (i.e., $1.2 \, mAh \, cm^{-2}$) and 25 °C.

## Results

**Design and fabrication of the BNQS electrode.** The structural design and fabrication procedure of the BNQS electrode are schematically represented in Fig. 1a, together with a photograph and cross-sectional scanning electron microscopy (SEM) image. A UV-curable gel electrolyte precursor was prepared by combining a liquid electrolyte (1 M $LiPF_6$ in ethylene carbonate (EC)/propylene carbonate (PC) = 1/1 (v/v)) with an ethoxylated trimethylolpropane triacrylate (ETPTA) monomer at a composition ratio of liquid electrolyte/ETPTA monomer of 85/15 (w/w). Subsequently, the UV-curable gel electrolyte precursor was mixed with NCM811 particles and carbon black conductive additives without using any processing solvents such as NMP, producing an electrode paste (NCM811/carbon black/gel electrolyte precursor = 75/5/20 (w/w/w)). The obtained electrode paste showed thixotropic fluid behavior (Supplementary Fig. 1), which is suitable for a stencil-printing process[33,34].

The electroconductive mat was fabricated by concurrent electrospraying (for SWCNT suspension solution)/electrospinning (for polyetherimide (PEI)-trimethylolpropane propoxylate triacrylate (TPPTA) mixture solution, PEI/TPPTA = 70/30 (w/w)), followed by thermal curing of TPPTA at 90 °C for 3 h, in which the hydrophobic TPPTA polymer network was added to maintain the structural stability of the resulting electroconductive mat upon contact with electrode slurries (Supplementary Fig. 2). The fabrication procedure of the electroconductive mat and its chemical structures are depicted in Supplementary Fig. 3a. The intermolecular π–π stacking interactions[26,35] between the aromatic rings of PEI and SWCNT allowed the conformal wrapping of the PEI–TPPTA fibers by the SWCNT (Supplementary Fig. 3b). The obtained electroconductive mat (thickness ~11 μm and a density of about $0.4 \, mg \, cm^{-2}$) showed an electronic conductivity of $25 \, S \, cm^{-1}$ (Supplementary Fig. 3c). Meanwhile, a large-sized electroconductive mat ($=10 \times 10 \, cm^2$) was fabricated to demonstrate the upscaling feasibility of the material production (Supplementary Fig. 3d).

On top of the electroconductive mat that can act as an alternative porous current collector replacing the conventional metallic-foil one, the above-prepared electrode paste was stencil-printed, followed by UV curing, producing a BNQS unit electrode (areal-mass (solely including NCM811, if not specified) loading ~$10 \, mg \, cm^{-2}$). The optimal thicknesses (40 ~ 60 μm) of active layers in the unit electrodes were determined by analyzing the electronic conductivity of the unit electrodes as a function of thickness of active layer (Supplementary Fig. 4). Subsequently, the stencil-printing-based electrode fabrication step was repeated, together with insertion of the electroconductive mats in the through-thickness direction, until reaching the mass-loading value of interest (Fig. 1a). After pressing (5 MPa) to ensure intimate

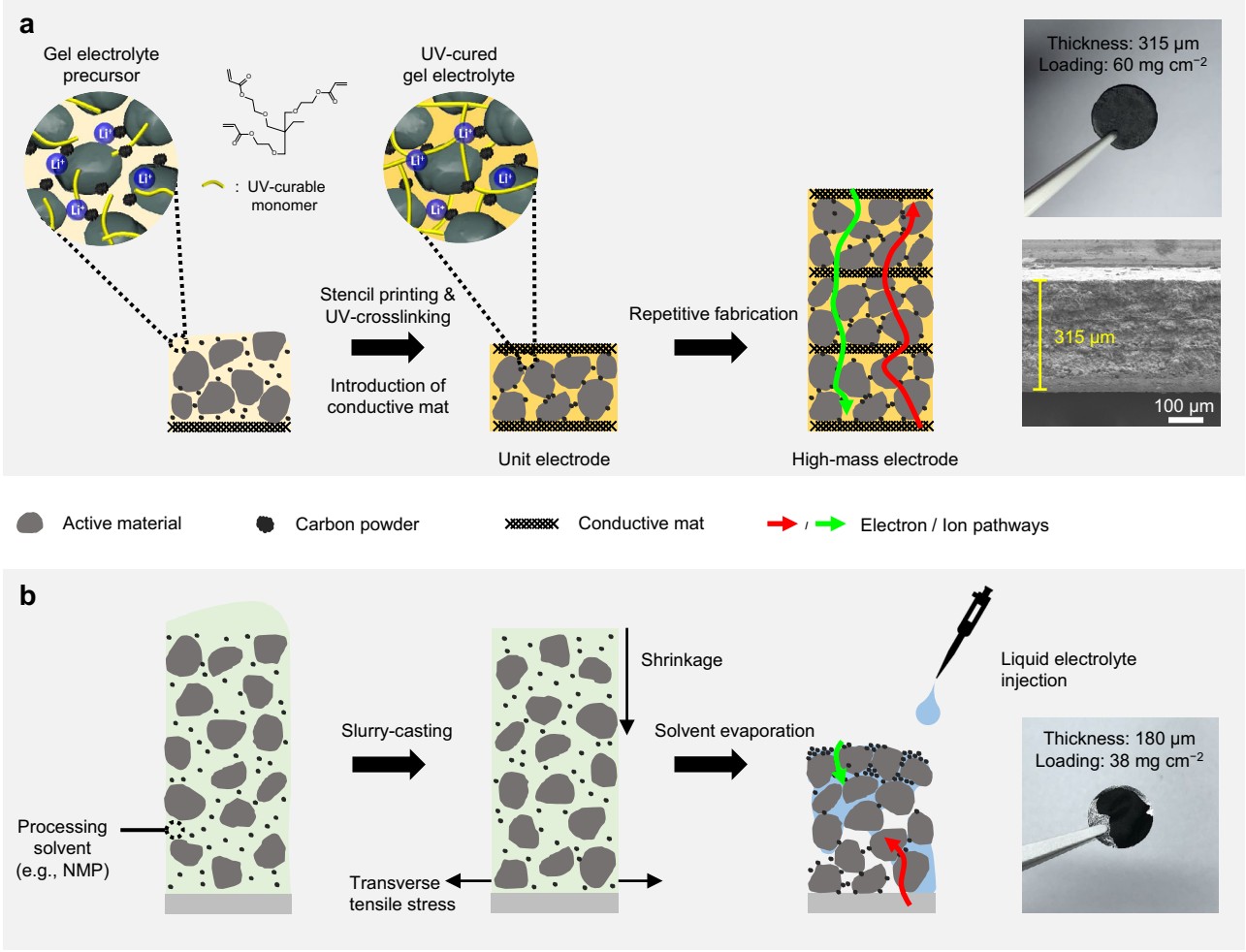

**Fig. 1 Schematic representation of the structural design and fabrication procedure of high-mass-loading electrodes. a** BNQS electrode, along with its photograph and cross-sectional SEM image. **b** Control electrode fabricated by means of a slurry-cast method.

interfacial contact between the electrode components, a metal-current-collector-free, electroconductive-mat/gel-electrolyte-embedded BNQS electrode was obtained (Supplementary Fig. 5). The basic information of the electroconductive-mat interlayers inserted in the BNQS electrodes was provided as a function of areal-mass-loading (Supplementary Table 1). We note that the BNQS electrode already includes 20 wt.% gel electrolyte (comprising the UV-cured ETPTA polymer and 1 M LiPF$_6$ in EC/PC electrolyte solution) besides NCM811 and carbon black additives, in which the gel electrolyte can be comparable to a binder and a liquid electrolyte in conventional LIB electrodes. The contents of binders and liquid electrolytes in conventional LIB electrodes are generally in the range of 15–20 wt.%[36], which are not significantly different from the gel electrolyte content of the BNQS electrode. Excluding the liquid electrolyte from the gel electrolyte leads to an electrode composition of NCM811/carbon black/ETPTA (this latter acting as the binder) = 90.5/6/3.5 (w/w/w), revealing that the NCM811 content in the BNQS electrode appears comparable to those of previously reported electrodes[25]. Meanwhile, a control electrode was fabricated by casting a slurry (NCM811/carbon black additive/polyvinylidene fluoride (PVdF) binder = 92/4/4 (w/w/w) in NMP as a processing solvent) on an aluminum (Al) current collector and followed by roll-pressing to equally match its thickness and areal-mass-loading with those of the BNQS electrode. In comparison with the BNQS electrode, the slurry-cast electrode failed to reach a thickness of higher than 180 µm because of unwanted crack formation and delamination during the drying of NMP (Fig. 1b)[9,10].

In addition to the thick electrode thickness, another achievement of the BNQS electrode is the shortened time for cell aging. This aspect is a major hurdle for the development of practical and cost-effective batteries with high-mass-loading electrodes. Based on electrochemical impedance spectroscopy measurements, we studied the bulk resistance ($R_{bulk}$, ionic resistance in the bulk electrolyte) using a symmetric blocking cell configuration equipped with electrodes with different mass loadings that are prepared via standard slurry-coated or stencil-printed methods. The slurry-cast electrode exhibited variance of $R_{bulk}$ with aging time, and it is more pronounced for the electrodes with higher mass loading (Supplementary Fig. 6a, b). In sharp contrast, the BNQS electrode showed a shorter aging time owing to the preinclusion of gel electrolytes (Supplementary Fig. 6c, d). Regardless of areal-mass-loading values (even at 36 mg cm$^{-2}$), a negligible change in the $R_{bulk}$ was observed with aging time.

**Structural analysis of the BNQS electrode.** The 3D microstructures of the BNQS and slurry-cast electrodes were investigated using focused-ion-beam nanotomography and associated 3D reconstruction techniques. For a fair comparison between the two electrodes, the electrode thickness was set to 160 µm (areal-mass-loading of 36 mg cm$^{-2}$), in which the slurry-cast electrode as well as the BNQS electrode maintained its structural integrity.

The slurry-cast electrode (shown in Fig. 2a) showed a nonuniform distribution of carbon powders (shown in yellow)

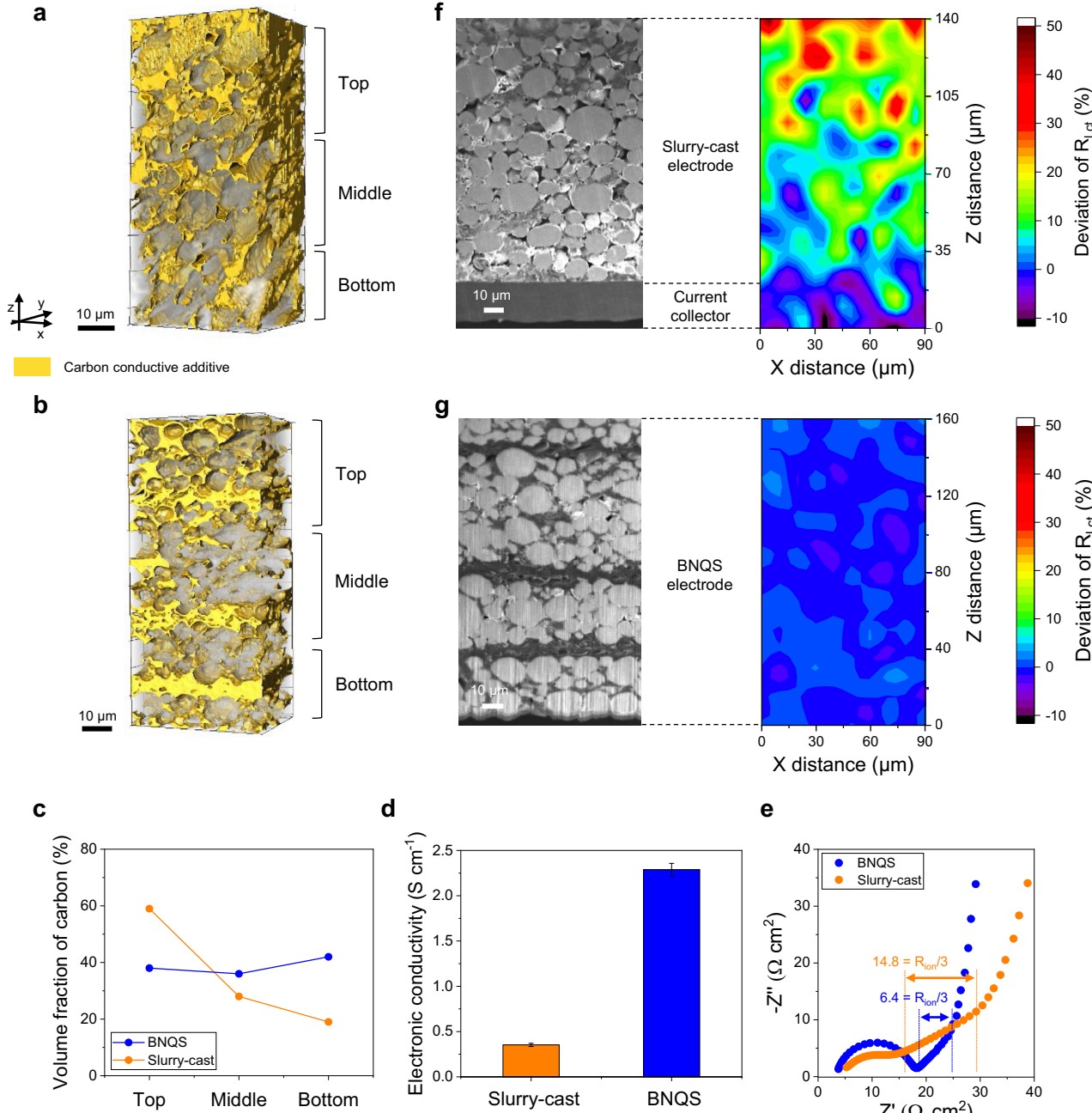

**Fig. 2 Structural analysis of the BNQS electrode (versus slurry-cast electrode).** 3D microstructural analysis, focusing on carbon (colored in yellow) distribution, of **a** slurry-cast and **b** BNQS electrodes, in which the electrode thickness was set as 160 μm (areal-mass-loading of 36 mg cm$^{-2}$) to make a fair comparison between the two electrodes. **c** Volume fraction of carbon in each section of the electrode along its z-axis direction. **d** Electronic conductivity of the electrodes. **e** Nyquist plots obtained by a symmetric cell configuration at 0% SOC. Cross-sectional SEM images and corresponding localized charge transfer resistance ($R_{l,ct}$) obtained from LEIS analysis of **f** slurry-cast and **g** BNQS electrodes.

in the through-thickness (z-axis) direction. Larger amounts of carbon powder were aggregated in the top layer than in the bottom layer. During the drying step of the electrodes, processing solvents evaporate predominantly from the electrode slurry surface, thus causing a nonuniform distribution of lightweight electrode components, such as carbon conductive powders[32]. As a result, the slurry-cast electrode failed to form well-interconnected electron conduction networks.

By comparison, the carbon powders were uniformly dispersed throughout the entire BNQS electrode (Fig. 2b). Because the BNQS electrode was fabricated without any processing solvents

owing to the presence of a gel electrolyte precursor in the electrode paste, the BNQS electrode did not undergo drying steps. This eliminates the issues associated with the solvent-drying-triggered random distribution of electrode components. This result was verified by quantitatively analyzing the volume fraction of carbon in each section of the electrode along its z-axis direction. In comparison with the slurry-cast electrode, the BNQS electrode showed an even distribution of carbon powders along its z-axis direction (Fig. 2c). This result was confirmed by examining the electronic conductivity of the electrodes by the four-point probe technique, in which the electronic conductivity

of the BNQS electrode was measured after removing the embedded gel electrolyte by solvent (dimethyl carbonate) rinsing followed by vacuum drying. The BNQS electrode, owing to its homogeneously distributed carbon powders and the electroconductive mats, exhibited higher electronic conductivity than the slurry-cast cathode (Fig. 2d).

In addition to the electron conduction networks mentioned above, ion conduction channels play an important role in achieving uniform redox reactions in high-mass-loading electrodes. To elucidate the ion transport phenomena inside the BNQS electrode, electrochemical impedance spectroscopy (EIS) measurement was conducted using a symmetric cell (electrode|separator|electrode) at 0% state of charge (SOC) (Fig. 2e). The Nyquist plot and corresponding curves fitted by a transmission line equivalent circuit model (TLM)[18,37] were provided in Supplementary Fig. 7. Ogihara et al.[38,39] reported that the projection of a slope (observed in the low-frequency region of complex impedance plot) to a real axis, which is defined as ionic resistance ($R_{ion}$)/3, reflects the ionic resistance inside the electrodes. The BNQS electrode showed a lower $R_{ion}$/3 (6.43 $\Omega$ cm$^2$) than the slurry-cast electrode (14.81 $\Omega$ cm$^2$). Meanwhile, the gel electrolyte inside the BNQS electrode provided slightly lower ionic conductivity (2.73 mS cm$^{-1}$) than the liquid electrolyte (4.90 mS cm$^{-1}$) inside the slurry-cast electrode (Supplementary Fig. 8). Such inconsistency between the ionic resistance inside the electrodes and the bulk ionic conductivity of the electrolytes reveals that the ion transport phenomena inside electrodes could be more influenced by their structural effect (i.e., homogeneity of ion conduction channels)[40]. As shown in Fig. 2a–c, the BNQS electrode provided the uniform distribution of components along the through-thickness direction compared to the slurry-cast electrode. For the slurry-cast electrode, a relatively dense structure was formed in the top layer due to the severe aggregation of carbon conductive additives, thus hindering the access of Li ion into the electrode[41]. In contrast, the BNQS electrode exhibited the homogeneous distribution of its components, beneficially contributing to the formation of highly interconnected ion conduction channels in the through-thickness direction[42].

This advantageous effect of the BNQS electrode on the electron/ion transport phenomena was further elucidated by carrying out EIS measurements and analysis of Li metal cells (NCM811 cathode ∥ Li metal anode) at 100% SOC. The BNQS electrode showed a lower resistance than the slurry-cast electrode (Supplementary Fig. 9). However, these conventional electrochemical characterization techniques are not sufficient for quantitatively identifying redox reactions in specific electrode spots because they show the averaged electrochemical response[43]. To address this challenging issue, the local electrochemical impedance spectroscopy[43] (LEIS) technique was used to investigate the localized charge transfer resistance ($R_{l,ct}$) of electrodes along their z-axis direction. A microscale probe (resolution of 10 μm) was employed as a counter electrode, and the impedance of the slurry-cast and BNQS electrodes (cross section) was measured (details of the LEIS measurement are shown in Supplementary Fig. 10). The slurry-cast cathode (Fig. 2f) showed a nonuniform distribution of $R_{l,ct}$ along the z-axis direction. Additionally, $R_{l,ct}$ tended to increase with the distance from the metal current collector side, confirming the solvent-drying-triggered nonuniform distribution of carbon conductive powders. By comparison, lower $R_{l,ct}$ values and their homogeneous distribution were observed over the entire BNQS electrode (Fig. 2g). This comparison of $R_{l,ct}$ supports the speculation of the uniform electrochemical reaction of the BNQS electrode along its through-thickness direction.

**Electrochemical performance of the BNQS electrode.** Based on the understanding of the redox homogeneity of the BNQS

electrode described above, its electrochemical performance was investigated using 2032-type coin cells (NCM811 cathodes (thickness = 160 μm and areal-mass-loading = 36 mg cm$^{-2}$)∥ Li metal anodes (thickness = 100 μm)) were used, in which a N (negative electrode capacity)/P (positive electrode capacity) ratio was 2.8. The electrolyte amount represented by a ratio of electrolyte mass to cell capacity (i.e., E/C) in the cell was set as 2.3 g Ah$^{-1}$ by considering the preferred amount of electrolyte (<3.0 g Ah$^{-1}$ for high-energy-density cells[44,45]). Many previous studies on high-mass-loading electrodes, unfortunately, did not mention electrolyte contents and just a few works reported the values (E/C ratio > 10 g Ah$^{-1}$, Supplementary Table 2)[44]. Such large amount of electrolytes in the electrodes may result in the unwanted loss of cell energy densities. In the coin cell with the BNQS cathode (already including the gel electrolyte of 1.1 g Ah$^{-1}$), liquid electrolyte of 1.2 g Ah$^{-1}$ (1 M LiPF$_6$ in EC/PC = 1/1 (v/v) with 10 wt.% FEC and 1wt.% vinylene carbonate (VC)) was added to electrochemically activate Li metal anode. For a control cell with the slurry-cast cathode, the same amount of liquid electrolyte (2.3 g Ah$^{-1}$, corresponding to 1.1 + 1.2 g Ah$^{-1}$) was injected to activate the cathode, and Li metal anode.

The charge/discharge rate capabilities of the BNQS and slurry-cast cathodes with the same areal-mass-loading of 36 mg cm$^{-2}$ were compared, in which the charge/discharge current rates varied from 0.05 (0.35 mA cm$^{-2}$) to 0.5 C (3.5 mA cm$^{-2}$). Under the high areal-mass-loading, the BNQS cathode enables the delivery of a specific discharge capacity of 191 mAh g$_{NCM811}^{-1}$ at 0.05 C, which almost reached the theoretical capacity (~192 mAh g$_{NCM811}^{-1}$)[46] of NCM811 (Fig. 3a). The slurry-cast cathode, however, failed to realize the theoretical capacity of NCM811, mainly because of the poorly developed ion/electron conduction channels (shown in Fig. 2), yielding a discharge capacity of 168 mAh g$_{NCM811}^{-1}$ at 0.05 C (Fig. 3b). Notably, the difference in the areal capacity between the BNQS and slurry-cast cathodes increased with increasing the current rate and areal-mass-loading (Fig. 3c), demonstrating the advantageous effect of the BNQS cathode on the redox kinetics of cells.

To verify the improved rate capability of the high-mass-loading BNQS cathode, galvanostatic intermittent titration technique (GITT) measurements and analysis was conducted during the charging/discharging of cells. Figure 3d shows that the BNQS cathode effectively mitigated the rise in cell polarization upon repeated current stimuli (applied at a C rate of 0.05 C (corresponding to 0.35 mA cm$^{-2}$) and interruption time between the pulses of 10 min), wherein the obtained internal cell resistances ($R_{internal}$) were provided as a function of SOC and depth of discharge (DOD). These GITT results were collected to calculate the Li ion diffusion coefficients ($D_{Li+}$) of the two cathodes (see Supplementary Fig. 11 and Table 3 for more calculation details[21]). The BNQS cathode (1.57 × 10$^{-8}$ cm$^2$ s$^{-1}$) showed a higher diffusion coefficient than the slurry-cast cathode (5.45 × 10$^{-9}$ cm$^2$ s$^{-1}$), which is consistent with the structural characterization shown in Fig. 2.

In addition to the GITT analysis, the overpotential distribution of the BNQS cathode in the through-thickness direction was examined using an in situ measurement of the cathode potential. For this characterization, a NCM811 ∥ Li pouch cell (Supplementary Fig. 12a) was prepared based on the results of previous studies[26,47]. During galvanostatic discharging at a C rate of 0.05 C (corresponding to 0.35 mA cm$^{-2}$), the voltage difference between the top and bottom sides of the cathodes was monitored (Supplementary Fig. 12b). At a discharge voltage of 3.7 V, the BNQS cathode showed a smaller voltage difference (ΔV ~ 8 mV) than that (ΔV ~ 15 mV) of the slurry-cast cathode.

Meanwhile, to further highlight the redox kinetics of the BNQS cathode, its charge/discharge behavior at various operating temperatures was investigated (Supplementary Fig. 13). Under

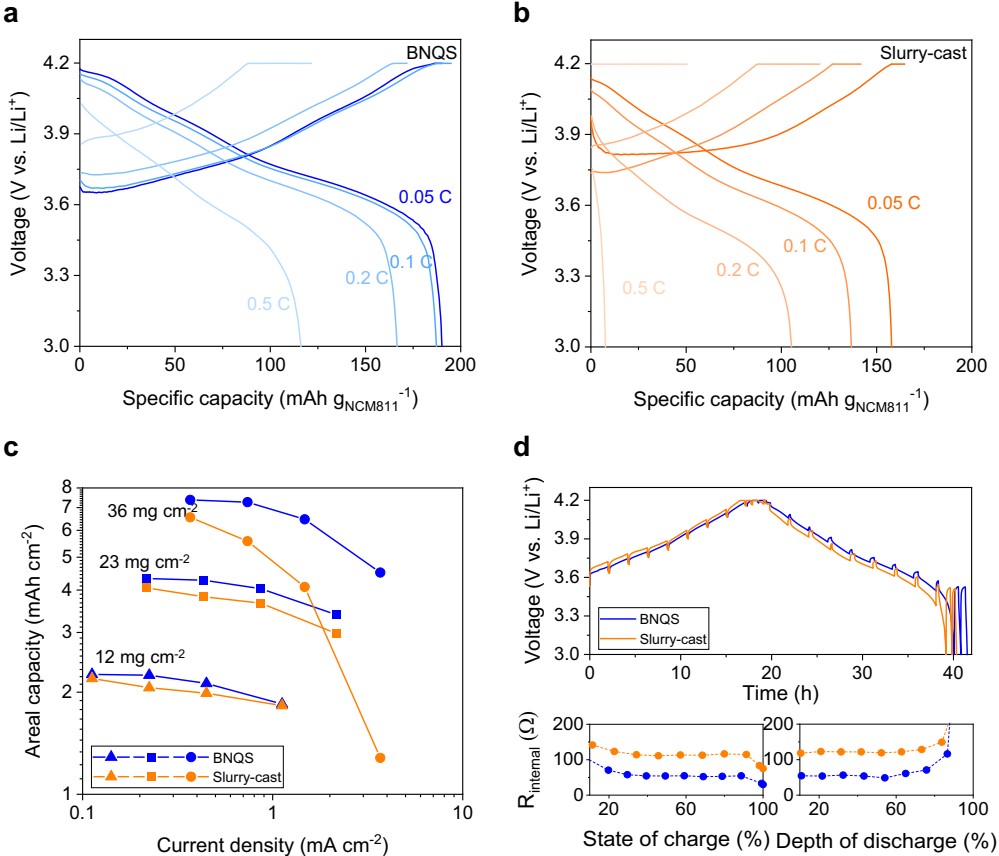

**Fig. 3 Electrochemical performance of the BNQS cathode (versus slurry-cast cathode).** Charge/discharge profiles of **a** slurry-cast and **b** BNQS cathodes (areal-mass-loading of 36 mg cm$^{-2}$) under varied charge/discharge current rates (0.05 C/0.05 C (=0.35 mA cm$^{-2}$) – 0.5 C/0.5 C (=3.5 mA cm$^{-2}$)). **c** Areal capacity of the cathodes with different areal-mass-loading (varying from 12 to 36 mg cm$^{-2}$) as a function of current density. **d** (top) GITT profiles upon repeated current stimuli at charge/discharge current density of 0.05 C/0.05 C (=0.35 mA cm$^{-2}$) and (bottom) internal cell resistance (R$_{internal}$) as a function of SOC and DOD.

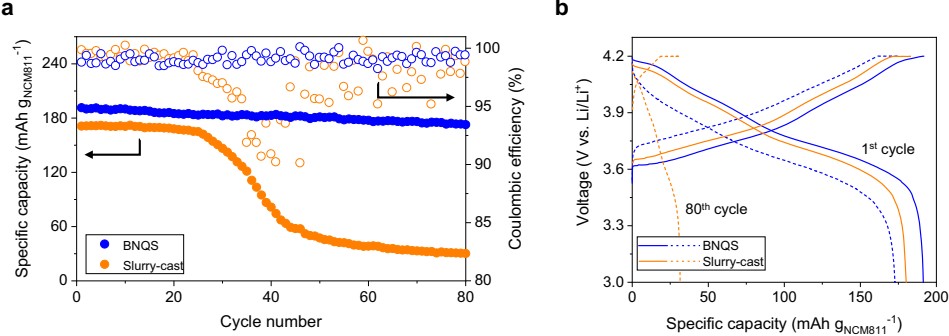

**Fig. 4 Cycling performance of the BNQS cathode (versus slurry-cast cathode).** **a** Cycling performance at charge/discharge current density of 0.05 C/0.1 C (=0.35 mA cm$^{-2}$/0.7 mA cm$^{-2}$) and voltage range of 3.0 –4.2 V under a limited amount of electrolyte (E/C ratio of 2.3 g Ah$^{-1}$), in which the coin cell containing the cathode with an areal-mass-loading of 36 mg cm$^{-2}$. **b** Charge/discharge profiles of the cells at 1 st and 80th cycle.

the same areal-mass-loading of 36 mg cm$^{-2}$, the BNQS cathode showed the higher discharge capacities than the slurry-cast cathode over a wide range of temperatures (varying from −10 to 60 °C) examined herein.

**Cycling performance of the BNQS electrode.** In addition to the improved redox kinetics described above, the NCM811 ‖ Li coin cell with the BNQS cathode showed higher capacity retention with cycling (90% after 80 cycles at a charge/discharge C rates of 0.05 C/0.1 C corresponding to 0.35 mA cm$^{-2}$ and 0.7 mA cm$^{-2}$,

respectively) than the control cell with the slurry-cast cathode (16% after 80 cycles at the same cycling conditions) (Fig. 4a, b), in which the BNQS and slurry-cast cathodes had the same areal-mass-loading of 36 mg cm$^{-2}$ (leading to the same N/P ratio of 2.8). This result underscores the viability of the BNQS cathode in improving the cycling performance.

To elucidate this cyclability difference between the two cathodes, a postmortem analysis of the electrodes harvested from cycled NCM811 ‖ Li coin cells was conducted after the cycling test. During cycling, heavy metal ions (e.g., Ni$^{2+}$) of cathode

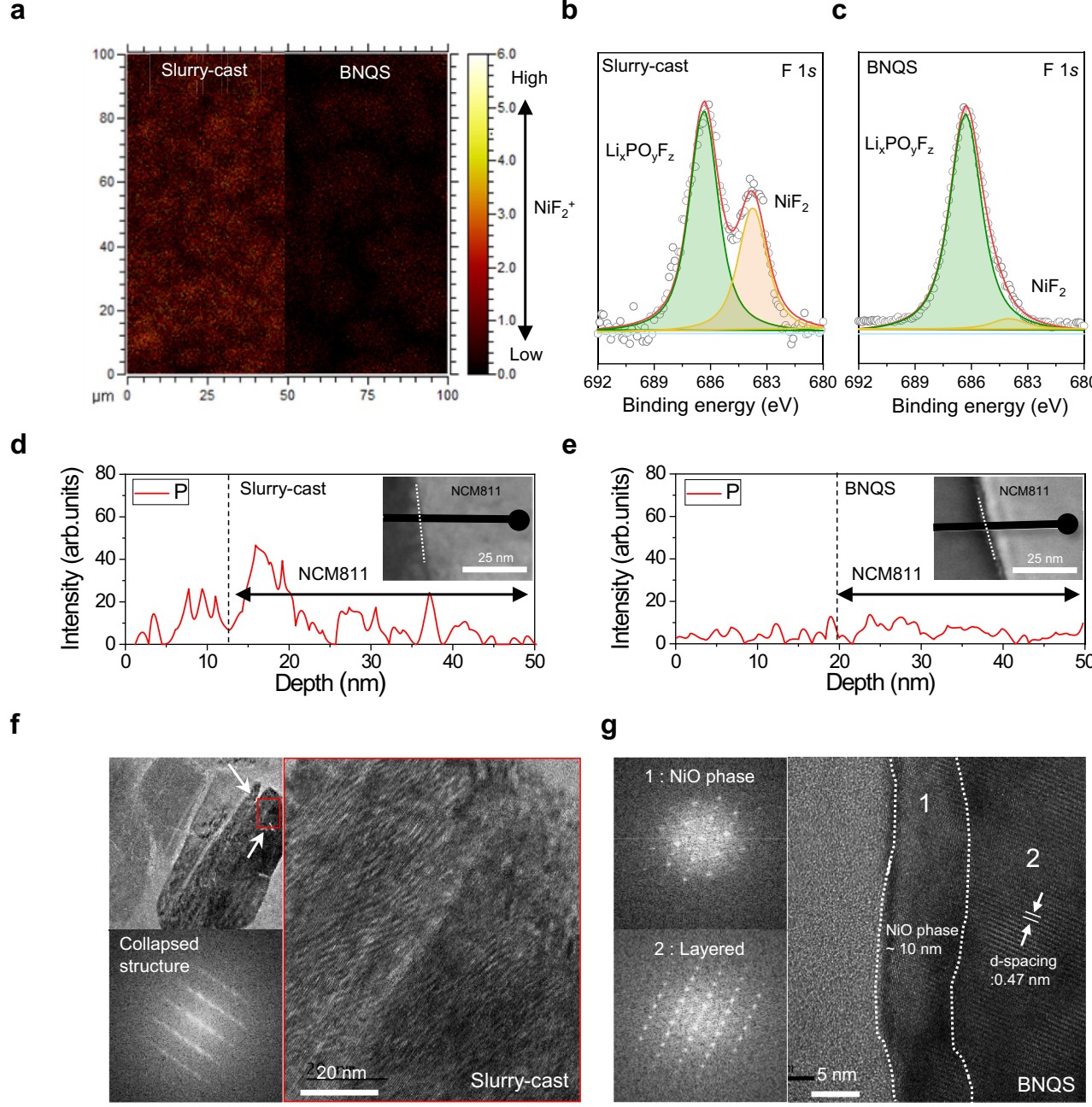

**Fig. 5 Post-mortem analysis of the cycled BNQS cathode (versus slurry-cast cathode). a** TOF-SIMS mapping images of the $NiF_2^+$ byproducts formed on the surface of the cathodes. XPS F 1s spectra, focusing on a characteristic peak assigned to $NiF_2$ (684.6 eV) byproducts of **b** slurry-cast and **c** BNQS cathodes. EDS line intensities for P element along the depth of NCM811 particles of **d** slurry-cast and **e** BNQS cathodes. HR-TEM images with fast Fourier transform patterns of the NCM811 particles of **f** slurry-cast and **g** BNQS cathodes.

materials tend to dissolve into electrolytes and are then deposited on anode materials, resulting in the formation of passivation layers[48,49]. An inductively coupled plasma mass spectroscopy (ICP-MS) analysis (Supplementary Fig. 14) revealed that the Li metal contamination by Ni deposition was significantly mitigated at the BNQS cathode compared with the slurry-cast cathode. This result indicates that the gel electrolyte inside the BNQS cathode effectively suppressed the dissolution of $Ni^{2+}$ from the NCM811. In addition, the surface of the cycled NCM811 particles was investigated. The time-of-flight secondary ion mass spectroscopy (TOF-SIMS) mapping images (Fig. 5a) showed that the formation of $NiF_2^+$ byproducts was suppressed on the BNQS cathode compared with the slurry-cast one. The X-ray photoelectron

spectroscopy (XPS) F1s spectra (Fig. 5b, c) revealed that a characteristic peak assigned to $NiF_2$ (684.6 eV) byproducts, which are generated by unwanted side reactions between NCM811 and electrolytes[31,50,51], was clearly observed at the slurry-cast cathode (Fig. 5b). These spectroscopic results demonstrate that the gel electrolyte in the BNQS cathode played a crucial role in preventing side reactions with NCM811. This beneficial effect of the BNQS cathode was verified by energy dispersive X-ray spectroscopy (EDS) line analysis along the depth of the NCM811 particles (Fig. 5d, e). A negligible amount of P element (coming from the $LiPF_6$ salt of the liquid electrolyte solution) was observed at the BNQS cathode compared with the slurry-cast one. Liquid electrolytes tend to penetrate NCM811 particles, thus causing

degradation of the layered structure and formation of a NiO phase during cycling[51–53]. This problem triggers the anisotropic change in the lattice parameter and crack generation of NCM811 particles, resulting in severe capacity decay[31,48].

To further highlight the advantageous effect of the BNQS cathode, the structural stability of the cycled NCM811 particles was characterized using high-resolution transmission electron microscopy (HR-TEM) analysis. The cycled NCM811 particles of the slurry-cast cathode showed a collapsed and disordered structure (Fig. 5f). The cycled NCM811 particles, because of their interfacial side reactions with electrolytes and nonuniform redox reactions during cycling, suffer from structural disruption, resulting in capacity loss with cycling[51–53]. In contrast, the cycled NCM811 particles of the BNQS cathode showed the uniform formation of a thin NiO phase (~10 nm) on its surface without cracks and disruption (Fig. 5g). Moreover, the layered structure having d-spacing (~ 0.47 nm) of the cycled NCM811 particles was stably preserved in the BNQS cathode (region 2 of Fig. 5g and Supplementary Fig. 15). This result demonstrates that the gel electrolyte inside the BNQS cathode beneficially contributed to the structural stability of the NCM811 particles during cycling.

In addition, the BNQS cathode showed an improvement in thermal stability. Differential scanning calorimetry (DSC) thermograms of the cathodes (charged to 4.2 V) showed that the UV-cured gel electrolyte in the BNQS cathode significantly decrease the interfacial exothermic reaction with the NCM811 compared to the liquid electrolyte in the slurry-cast cathode (Supplementary Fig. 16 and Supplementary Table 4). Indeed, the BNQS cathode shows a slightly higher exothermic temperature peak (2nd $T_{Peak}$) compared to the slurry-cast cathode (i.e., 229.9 °C vs 217.3 °C). In contrast, the exothermic heat ($\Delta H_{exothermic}$) of the BNQS cathode is more than four-fold lower than the one of the slurry-cast cathode (i.e., 219.3 J g$^{-1}$ vs 945.8 J g$^{-1}$).

**Enabling practical high-energy-density Li metal cells by the BNQS cathodes**. The BNQS cathodes showed stable charge/discharge behavior a voltage range of 3.0–4.2 V over the entire range of areal-mass-loadings explored herein (Supplementary Fig. 17), in which the BNQS cathodes were coupled with the Li metal anode (100 μm). The BNQS cathode with an areal-mass-loading of 60 mg cm$^{-2}$, of which cross-sectional morphology was shown in Supplementary Fig. 18, reached a (cathode sheet-based) areal capacity of 11.6 mAh cm$^{-2}$. The $D_{Li+}$ of the BNQS cathode was measured to be $1.63 \times 10^{-8}$ cm$^2$ s$^{-1}$ (Supplementary Fig. 19 and Supplementary Table 5), which is comparable to that of the BNQS cathode with the areal-mass-loading of 36 mg cm$^{-2}$ ($D_{Li+} = 1.57 \times 10^{-8}$ cm$^2$ s$^{-1}$, Supplementary Fig. 11). The discharge rate capability of the BNQS cathode with the areal-mass-loading of 60 mg cm$^{-2}$ was provided in Supplementary Fig. 20. The discharge capacity at a C rate of 0.5 C (corresponding to 5.8 mA cm$^{-2}$) was relatively low, which could be due to large amount of current applied and long tortuous pathways of ions and electrons. Future studies will be devoted to further enhancing the rate capability via fine-tuning of electrode structure and materials chemistry. To increase the cell energy densities further, the charge cut-off voltage was raised to 4.4 V. The higher charge voltage led to an increase in the areal capacities, eventually reaching an areal capacity of 12.3 mAh cm$^{-2}$ (Fig. 6a) which leads to a N/P ratio of 1.6 in the cell.

Realizing the theoretical capacities of electrode materials is a prerequisite to producing high-energy-density Li metal cells, and it has remained a formidable challenge in high-mass-loading electrodes. The BNQS cathodes achieved the full realization of the theoretical specific capacity of NCM811 over the entire range of areal-mass-loadings (Fig. 6b and Supplementary Fig. 21).

Previous works on high-mass-loading cathodes reported an increase in the absolute values of areal-mass-loadings; however, most failed to reach theoretical specific capacities of cathode materials (Fig. 6b and Supplementary Table 6), exhibiting their limitation in enabling uniform redox reaction in high-mass-loading electrodes.

The aforementioned efficiency of capacity utilization in the electrodes crucially affects the cell energy densities. The specific energy/energy density (without packaging) of the BNQS cathode-containing coin cell (404 Wh kg$^{-1}$/1025 Wh L$^{-1}$) were compared with those of previously reported high-mass-loading cathode systems (Fig. 6c, Supplementary Fig. 22, and Supplementary Table 6), in which the specific energies/energy densities were estimated without including packaging substances (see Supplementary Table 7 for calculation details). The highest specific energies/energy densities (without packaging) achieved by the BNQS cathode having an areal capacity of 12.3 mAh cm$^{-2}$ exceeded those of previous studies, demonstrating the viability of the BNQS cathode approach in enabling the high-energy-density cell.

Furthermore, a single-side-electrode pouch cell (26 × 24 mm$^2$ in size, Supplementary Fig. 23) with the BNQS cathode was fabricated and its performance was investigated at a voltage range of 3.0–4.4 V to explore the feasibility of this high-mass-loading electrode concept. The pouch-type cell was composed of the BNQS cathode (areal capacity of 12.1 mAh cm$^{-2}$)|| Li metal anode (areal capacity of 20 mAh cm$^{-2}$), in which the N/P ratio was 1.6. The amount of electrolyte in the cell (including the gel electrolyte in the BNQS cathode) was limited as 2.3 g Ah$^{-1}$ to practically evaluate the cell[44,54]. The obtained pouch-type cell showed an initial discharge capacity of 203 mAh g$^{-1}$ (Fig. 6d), exhibiting almost full utilization of the theoretical specific capacity (205 mAh g$^{-1}$)[46] of NCM811 in the BNQS electrode. In addition, the specific energy/energy density (with packaging) of the pouch-type cell were estimated to reach 321 Wh kg$^{-1}$/772 Wh L$^{-1}$ at SOC 0% (Supplementary Table 8). The pouch-type cell showed stable cycling performance (Fig. 6e), underscoring the electrochemical viability of the high-mass-loading BNQS cathodes. Meanwhile, no significant difference in the electrode thickness was observed before and after the cycling test (Supplementary Fig. 24), indicating the structural stability of the BNQS cathode with cycling. Intriguingly, the areal capacity of the pouch-type cell was almost recovered to an initial value, after the cycled Li metal anode was replaced[55,56] with a fresh one after 65 cycles (Fig. 6e). This result may indicate that the BNQS cathode is not a crucial cause of the cycling fading of the pouch-type cell.

**Discussion**

In summary, we presented the redox-homogeneous BNQS electrodes as an approach for high-mass-loading electrodes. Mixing UV-curable gel electrolyte precursors (acting as a processing solvent) with NCM811 particles and carbon additives during the electrode fabrication allowed the removal of solvent-drying steps, eventually resulting in the uniform distribution of electrode components throughout the entire region of the BNQS electrode, along with the formation of percolated ion-conduction networks. The electroconductive-mat interlayers based on the SWCNT-wrapped PEI-TPPTA nanofibers in the BNQS electrodes ensured facile electron conduction and eliminated heavy metallic-foil current collectors. The BNQS cathode reached a high areal-mass-loading level (60 mg cm$^{-2}$, corresponding to an areal capacity of 12.3 mAh cm$^{-2}$), which was difficult to achieve with conventional slurry-cast cathodes. More notably, the theoretical specific capacity of NCM811 was almost realized over a wide range of mass loadings, demonstrating the almost full utilization of the

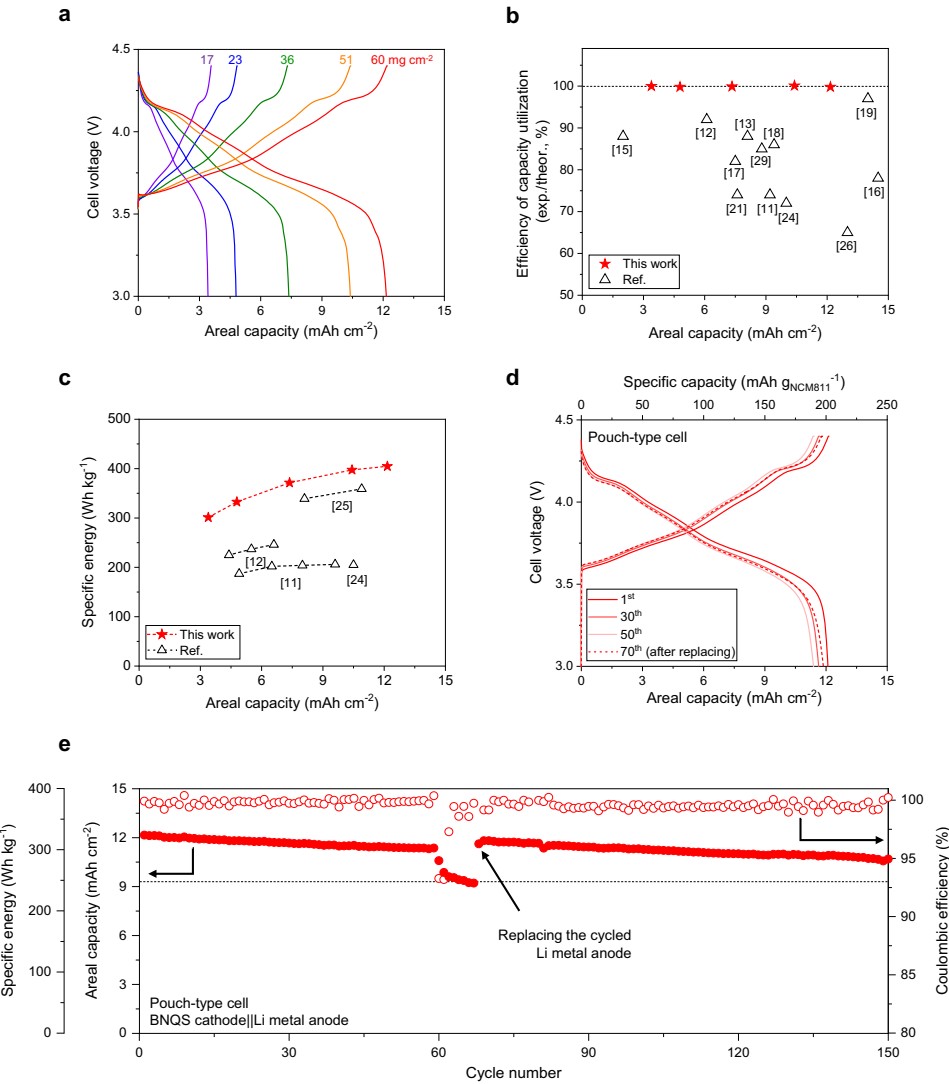

**Fig. 6 Testing of Li metal cells with high-mass-loading cathodes. a** Charge/discharge profiles of the coin cells (BNQS cathode || Li metal anode (100 μm corresponding to an areal capacity of 20 mAh cm$^{-2}$)) as a function of areal-mass-loading of the BNQS cathodes at a voltage range of 3.0–4.4 V and charge/discharge current density of 0.05 C/0.1 C. **b** Realization of theoretical specific capacities of cathode materials as a function of areal capacity (BNQS cathodes versus previously reported cathodes). **c** Specific energies of cells as a function of areal capacity (BNQS cathodes versus previously reported cathodes), in which the cell weight was estimated by considering those of cathodes, anodes, separators, and electrolytes. **d** Charge/discharge profiles of the single-side pouch-type cell (26 × 24 mm$^2$ in size) composed of BNQS cathode (areal capacity of 12.1 mAh cm$^{-2}$)|| Li metal anode (areal capacity of 20 mAh cm$^{-2}$) at charge/discharge current rate of 0.05 C/0.1 C (=0.6 mA cm$^{-2}$/1.2 mA cm$^{-2}$) and voltage range of 3.0–4.4 V. **e** Cycling performance of the pouch-type cell. The specific energy (with packaging) of the pouch-type cell was estimated by considering its entire cell weight. The cell was reassembled by replacing the cycled Li metal anode with a fresh one after 65 cycles. The cell performance was conducted at 25 °C.

electrochemical activity of NCM811 in the BNQS cathode. The pouch-type Li metal cell (NCM811 (12.1 mAh cm$^{-2}$)|| Li (20.0 mAh cm$^{-2}$) corresponding to an N/P ratio of 1.6) exhibited a specific energy and energy density of 321 Wh kg$^{-1}$ and 772 Wh L$^{-1}$ (based on the total mass of the cell), respectively.

## Methods

**Fabrication of BNQS electrodes**. The electroconductive mat was fabricated by concurrent electrospinning (for polyetherimide-trimethylolpropane propoxylate triacrylate (PEI-TPPTA) nanofibers) and electrospraying (for SWCNT, TUBALL$^{TM}$) through two different nozzles. For the electrospinning, PEI powder/TPPTA monomer (= 70/30 (w/w)) were dissolved in N-methyl-2-pyrrolidone (NMP, Aldrich)/dimethylacetamide (DMAc) (= 25/75 (w/w)) with benzoyl peroxide (BPO) as a thermal-initiator. For the electrospraying, the SWCNT suspension (solid content = 1 wt.% in water/isopropyl alcohol (IPA) (=90/10 (w/w)) was used, in which 1 wt.% of polyvinylpyrrolidone (PVP) was incorporated as a dispersion agent. The detailed processing conditions were 11 kV with a feed rate of

3 μL min$^{-1}$ for the electrospinning and 14 kV with a feed rate of 60 μL min$^{-1}$ for the electrospraying. The obtained mat was thermally cured at 90 °C for 3 h and then roll-pressed through a speed of 1 m min$^{-1}$ at 80 °C for a press ratio of 10%, producing a self-standing electroconductive interlayer.

The paste of BNQS electrodes was composed of NCM811(LiNi$_{0.8}$Co$_{0.1}$Mn$_{0.1}$O$_2$, LG chem)/carbon black (super C65, LG Chem)/gel electrolyte precursor (= 75/5/20 (w/w/w)) without any processing solvents. The gel electrolyte precursor was prepared by mixing UV-curable ethoxylated trimethylolpropane triacrylate (ETPTA) monomer (Aldrich) (including 1 wt.% 2-hydroxy-2-methl-1-phenyl-1-propanone (HMPP) as a photo-initiator) and non-aqueous electrolyte solution (1M LiPF$_6$ in ethylene carbonate(EC)/propylene carbonate(PC) 1/1 (v/v)) at a composition ratio of 15/85 (w/w). To fabricate the BNQS electrode, the electrode paste was directly deposited into the above-prepared electroconductive-mat using a stencil-printing, in which the thickness of electrode active layer was adjusted by the thickness of a master mold. Then, another electroconductive-mat was placed on top of the printed electrode and followed by UV irradiation (Hg UV-lamp (Lichtzen)) with an irradiation peak intensity of 1260 mW cm$^{-2}$ for solidification above 5 cm apart from the electrode, yielding a BNQS unit electrode. This procedure was repeatedly conducted to reach the designed areal-mass-loading

value. After pressing (5 MPa) to ensure intimate interfacial contact between the electrode components, a metal-current-collector-free, electroconductive-mat/gel-electrolyte-embedded electrode was obtained. As a control sample for the comparative analysis, a slurry-cast electrode was fabricated by casting a slurry mixture (NCM811/polyvinylidene fluoride (PVdF)/carbon black = 92/4/4 (w/w/w) in NMP) on an Al foil. The casted electrode slurry was vacuum-dried at 120 °C for 12 h and pressed (10 MPa~20 MPa) in order to equally match its thickness and areal-mass-loading with those of the BNQS electrode. All fabrication of the electrodes was conducted in a dry room with a dew point of −50 °C and area of $3 \times 4$ m$^2$.

**Structural and physicochemical characterizations of BNQS electrodes**. Before any characterizations of the active material of the BNQS electrode, the gel electrolyte was removed by solvent (dimethyl carbonate (DMC), ≥ 99%, Soulbrain) rinsing followed by vacuum drying at 120 °C. The cross-sectional morphologies of BNQS and control electrodes were investigated using field emission SEM (S-4800, Hitachi). A dual-beam FIB-SEM (NX2000, Hitachi) was used to examine the cross-sectional microstructure of electrodes, in which argon (Ar) ion milling (Model 1040 Nanomill, Fischione) was performed to obtain sample specimens. The FIB 3D reconstruction was conducted with the aid of Image-Pro Premier 3D (Media Cybernetics). The electronic conductivity of electrodes was measured using the four-point probe technique (CMT-SR1000N, Advanced Instrument Tech). For the post-mortem analysis, the cells after 80 cycles were disassembled at the discharging state and rinsed with a solvent (DMC) to remove residual salts in an Ar-filled glovebox (oxygen and water were less than 0.1 ppm). The samples for the post-mortem analyses were transferred using the sealed pouch with inert gas. The inductively coupled plasma–mass spectroscopy (ICP–MS, ELAN DRC–II, Perkin Elmer) analysis was conducted to quantitatively estimate amount of deposited Ni on the Li metal anode. The chemcial change of cathode surface after cycling test was analyzed using time-of-flight secondary ion mass-spectroscopy (TOF-SIMS, ION TOF) with Bi$_3^{2+}$ gun (25 keV, 1 pA). The chemical change of cathode surface after the cycling test was analyzed using X-ray photoelectron spectroscopy (XPS, ThermoFisher) with focused monochromatized Al Kα radiation. To analyze the microstructure of NCM811 particles, cross-sectioned samples were thinned by using focused ion beam (FIB, Helios Nano Lab, FEI). The structural and chemical analysis of the thinned samples was conducted by high resolution transmission electron microscope (HRTEM, ARM300, JEOL). The energy-dispersive X-ray spectroscopy (EDS) line analysis was conducted using the same HRTEM. The interfacial exothermic reaction between the charged NCM811 and electrolyte was examined using differential scanning calorimetry (DSC, Q200, TA), in which the cells were charged to 4.2 V at a C rate of 0.05 C (0.35 mA cm$^{-2}$) and then disassembled in a glove box to obtain the charged NCM811.

**Electrochemical characterization of BNQS electrodes**. The ion conductivity of electrodes was estimated by electrochemical impedance spectroscopy (EIS) measurement and analysis of the symmetric cells (electrode|separator|electrode) at a frequency range from $10^{-2}$ to $10^6$ Hz and an applied amplitude of 10 mV using potentiostat/galvanostat (VSP classic, Bio-Logic). The localized electrochemical impedance spectroscopy (LEIS) area scans were measured using a scanning probe workstation (M470, Biologic) at a fixed frequency of 50 Hz and an applied amplitude of 100 mV with 10 μm spacing. To investigate a directional distribution of interfacial charge transfer resistance ($R_{l,ct}$), ion-milled electrodes were insulated except its joint between the Cu current collector and working electrode. The electrochemical performance of BNQS electrodes was characterized using a 2032-type coin cell or pouch-type cell (composed of BNQS cathode ‖ Li metal anode (thickness = 100 μm), electrolyte: 1 M LiPF$_6$ in EC/PC = 1/1 (v/v) with 10 wt.% FEC and 1wt.% vinylene carbonate (VC), electrolyte mass/cell capacity (E/C) ratio = 2.3 g Ah$^{-1}$ (= 1.1 g Ah$^{-1}$ of gel electrolyte in the cathode and 1.2 g Ah$^{-1}$ of injected electrolyte to electrochemically activate the separator and Li metal anode), if not specified. The single-side-electrode pouch cell (composed of BNQS cathode (areal capacity of 12.1 mAh cm$^{-2}$)‖ Li metal anode (areal capacity of 20 mAh cm$^{-2}$) was fabricated using an Al pouch film as a packaging substance in a dry room with a dew point of -50 °C and area of $3 \times 4$ m$^2$. The electrochemical performance of the pouch-type cell was evaluated after the initial formation for 2 cycles (at a charge/discharge C rate of 0.05 C (=0.6 mA cm$^{-2}$)) under a fixed pressure set as 500 kPa[57,58]. The galvanostatic intermittent titration technique (GITT) analysis was conducted with interruption time between each pulse of 10 min. The cell performance was investigated using a cycle tester (PNE Solution) in a chamber set as 25 °C at various charge/discharge conditions.

**Calculation of the specific energy and energy density**. The specific energies and energy densities (without packaging) of the cells were estimated based on the weight and volume of cathode (including electroconductive-mat interlayers and gel electrolyte), anode (including a current collector), separator, and electrolyte in the cell. Meanwhile, the specific energy and energy density (with packaging) of the pouch-type cell was estimated based on the experimentally measured weight and volume of the cell (including the pouch packaging film). Calculation details for the

energy densities (with and without packaging) are described in Supplementary Table 7 and 8.

## Data availability
The authors declare that the main data supporting the findings of this study are available within the article and its Supplementary information. Extra data are available on reasonable request from the corresponding author.

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

## Acknowledgements

This work was supported by the Basic Science Research Program (2021R1A2B5B03001615 and 2018M3D1A1058744) through the National Research Foundation of Korea (NRF) grant by the Korean Government (MSIT). This research was also supported by the Technology Innovation Program (20010960) funded by the Ministry of Trade, Industry & Energy (MOTIE, Korea) and the Yonsei University Research Fund of 2020-22-0536.

## Author contributions

J.H.K., J.M.K. and S.Y.L. designed this work. J.H.K. and J.M.K. performed the experimental characterization and electrochemical tests. J.M.K and N.Y.K participated in the fabrication of electroconductive mat. J.H.K. and S.K.C. designed the LEIS analysis. S.Y.L. supervised the overall project. All authors contributed to finalizing the manuscript.

## Competing interests

The authors declare no competing interests.
