## [Peer Review File · Nature Communications]

REVIEWER COMMENTS

Reviewer #1 (Remarks to the Author):

Comments to authors:

This manuscript by Kim et al. reports on design and construction of redox-homogeneous, quasi-solid-state high-mass-loading electrodes for high-energy-density lithium batteries. Using a layer-by-layer structural design, the authors were able to construct ultrathick electrodes with a thickness of up to 315 μm and a mass loading of 59 mg/cm^2 , representing one of the highest values ever reported. One interesting benefit of this structural design is the excellent transporting properties of both ions and electrons, which has been a long-standing challenge of the field of thick electrode design. Another benefit is the high mass and volume ratio of electroactive materials and low ratio of non-active parts such as electrolyte, conductive additive, and separator, addressing another challenge of the field – the porosity of thick electrode is generally too high compared with commercial counterparts that leads to low volumetric capacity and energy density of the final devices. The manuscript was well written with well-organized and presented data. More attractively, this manuscript points out a promising electrode design concept towards high-energy batteries. The referee believes this excellent work can spark many further studies in the broad materials, chemistry, and energy communities. I would like to recommend the publication of this interesting manuscript after some minor points addressed.

1. At the end of one sentence, one should not cite too many references. For example, more than 18 refs were cited here “Along with these material-based works, the design of high-mass-loading electrodes has recently garnered considerable attention as a facile and scalable architectural strategy.^{9–27}”, which is far too much. These refs are good and relevant, but I would suggest the authors to consider cite them more suitably.
2. How is the cost of the developed electrodes and devices? In real applications, cost is as important as performance, and sometime more sensitive than performance toward commercialization. Although the referee is aware that this is a fundamental study that focused more on the scientific aspects, but I would still suggest the authors to discuss on this point given that some readers may be interested to know its cost.
3. For cell-level energy density calculation, why packaging materials were not included. As the referee know, a battery needs to be sealed as a cell, either coin cell, pouch cell, or other types of packaging.
4. Regarding the electrode thickness and mass loading, some previous reported thick electrodes can reach up to millimeter and more than 80 mg/cm^2 . Is the current design able to reach similar values? And what are the limits for the thickness and mass loading using this design?
5. A mass loading of 36 mg/cm^2 is not the highest for slurry-casted electrode. Some previous works reported higher values. So please be caution in making such claim.
6. The controlled slurry-casted electrode should also be pressed to make better contact of electrode materials and reduce the porosity fore fairer comparison, as the layer-by-layer structured electrodes were pressed for these purposes.

Chaoji Chen
April 8, 2021

Comments to authors:

This manuscript by Kim et al. reports on design and construction of redox-homogeneous, quasi-solid-state high-mass-loading electrodes for high-energy-density lithium batteries. Using a layer-by-layer structural design, the authors were able to construct ultrathick electrodes with a thickness of up to 315 μm and a mass loading of 59 mg/cm^2 , representing one of the highest values ever reported. One interesting benefit of this structural design is the excellent transporting properties of both ions and electrons, which has been a long-standing challenge of the field of thick electrode design. Another benefit is the high mass and volume ratio of electroactive materials and low ratio of non-active parts such as electrolyte, conductive additive, and separator, addressing another challenge of the field – the porosity of thick electrode is generally too high compared with commercial counterparts that leads to low volumetric capacity and energy density of the final devices. The manuscript was well written with well-organized and presented data. More attractively, this manuscript points out a promising electrode design concept towards high-energy batteries. The referee believes this excellent work can spark many further studies in the broad materials, chemistry, and energy communities. I would like to recommend the publication of this interesting manuscript after some minor points addressed.

1. At the end of one sentence, one should not cite too many references. For example, more than 18 refs were cited here “Along with these material-based works, the design of high-mass-loading electrodes has recently garnered considerable attention as a facile and scalable architectural strategy.^{9–27}”, which is far too much. These refs are good and relevant, but I would suggest the authors to consider cite them more suitably.
2. How is the cost of the developed electrodes and devices? In real applications, cost is as important as performance, and sometime more sensitive than performance toward commercialization. Although the referee is aware that this is a fundamental study that focused more on the scientific aspects, but I would still suggest the authors to discuss on this point given that some readers may be interested to know its cost.
3. For cell-level energy density calculation, why packaging materials were not included. As the referee know, a battery needs to be sealed as a cell, either coin cell, pouch cell, or other types of packaging.
4. Regarding the electrode thickness and mass loading, some previous reported thick electrodes can reach up to millimeter and more than 80 mg/cm^2 . Is the current design able to reach similar values? And what are the limits for the thickness and mass loading using this design?
5. A mass loading of 36 mg/cm^2 is not the highest for slurry-casted electrode. Some previous works reported higher values. So please be caution in making such claim.
6. The controlled slurry-casted electrode should also be pressed to make better contact of electrode materials and reduce the porosity fore fairer comparison, as the layer-by-layer structured electrodes were pressed for these purposes.

Chaoji Chen

April 8, 2021

陈朝吉

Reviewer #2 (Remarks to the Author):

After carefully reading this manuscript, this reviewer found that there are some critical problems. So, this work is not suitable for publication.

- 1) Most of results are based on the coin cell (only one charge-discharge curve in figure 5D is based on pouch cell). The practice energy density of the cell is calculated on the cathode, anode, electrolyte and separator excluding weights of package of coin cell and Al/Cu foil films, which is not acceptable.
- 2) The results on the coin cell are impossible to scale up for pouch cell. That is why there are only one charge-discharge curve in figure 5D. It is better to show the cycle performance of pouch cell.
- 3) The gravimetric and volumetric energy density are based on the calculation method shown in Table S3. This is not acceptable. The practice energy densities should be based on the measured weight and volume of the cell.
- 4) Based on the composition of NCM811(LiNi_{0.8}Co_{0.1}Mn_{0.1}O₂)/carbon black/gel 370 electrolyte precursor (=75/5/20 (w/w/w)), the energy density of pouch cell should be difficult to reach 410Wh/kg.

Reviewer #3 (Remarks to the Author):

In the manuscript entitled "Redox-homogeneous, quasi-solid-state high-mass-loading electrodes for high-energy-density lithium batteries", the authors reported NCM811 based novel cathode with high mass loading for lithium ion battery with high energy density (> 400 Wh/kg). It is very interesting to see that the optimized NCM811 cathode exhibits high conductivity, homogenous microstructure, and hence excellent redox reactions with multilayer connected by the conductive mat. Although this study shows promise for the practical application of the golden NCM811, the authors are needed to address the following major issues before going forward for publication.

1. The main novel point of this work is the capability to fabricate high mass loading (59.3 mg/cm²) cathode with high electrochemical performance. However, the major results are based on the electrode with mass loading of 36 mg cm⁻², in order to compare with the conventional slurry based cathode. To demonstrate the capability, cathode with 59.3 mg/cm² should be also characterized by electrochemical tests such as discharge with varied rates; Li diffusion coefficient; cycling performance, etc., as well as SEM tests.
2. The electrode with 160 um has a mass loading of 36 mg cm⁻². Why does that with ~315 um has the mass loading to 59.3 mg cm⁻²? The fabrication process is not linear?
3. The specific discharge capacity of 191 mAh/g is based on the mass of active material only or the whole electrode? The gel electrolyte has a weight ratio of 20%.
4. In figure 2E, it is hard to directly attribute the better ion transport behavior of BNQS to the structural effect. The authors should provide more evidence about it. Moreover, why the BNQS electrode has a larger semi-circle? More explanations should be added.
5. The increase of the cut-off voltage of course will increase the capacity and energy density. However, it will also influence cycling performance. It should be consistent. It is not suitable to use 4.2 V to get long cycling life but use 4.4 V to obtain the high energy density. Meanwhile, only 80 cycles are very limited.
6. It is recommended to characterize the structure and morphology of the cathode after cycling.
7. Is Figure 5a based on coin cell or pouch cell? The data of 59.3 mg cm⁻² is almost the same with that in Figure 5d. It is not possible to get the same data from coin cell and pouch cell. The authors should compare them.

8. The cycling performance of pouch cells should be characterized to show the potential of the practical applications.

Reviewer #1

This manuscript by Kim et al. reports on design and construction of redox-homogeneous, quasi-solid-state high-mass-loading electrodes for high-energy-density lithium batteries. Using a layer-by-layer structural design, the authors were able to construct ultrathick electrodes with a thickness of up to 315 μm and a mass loading of 59 mg cm^{-2} , representing one of the highest values ever reported. One interesting benefit of this structural design is the excellent transporting properties of both ions and electrons, which has been a long-standing challenge of the field of thick electrode design. Another benefit is the high mass and volume ratio of electroactive materials and low ratio of non-active parts such as electrolyte, conductive additive, and separator, addressing another challenge of the field – the porosity of thick electrode is generally too high compared with commercial counterparts that leads to low volumetric capacity and energy density of the final devices. The manuscript was well written with well-organized and presented data. More attractively, this manuscript points out a promising electrode design concept towards high-energy batteries. The referee believes this excellent work can spark many further studies in the broad materials, chemistry, and energy communities. I would like to recommend the publication of this interesting manuscript after some minor points addressed.

1. At the end of one sentence, one should not cite too many references. For example, more than 18 refs were cited here “Along with these material-based works, the design of high-mass-loading electrodes has recently garnered considerable attention as a facile and scalable architectural strategy.⁹⁻²⁷”, which is far too much. These refs are good and relevant, but I would suggest the authors to consider cite them more suitably.

→ Thank you so much for the reviewer’s constructive comment. In response to the reviewer’s comment, the references were divided to specific themes and rearranged in the following sentences, respectively. For a better understanding, representative review papers on high-mass-loading electrodes were added in ref. [7-8].

[Revised manuscript]

“The promise of forthcoming smart portable electronic devices, electronic vehicles (EVs), drones, and the Internet of things has spurred the relentless pursuit of high-energy-density

lithium (Li) batteries.¹⁻³ Many previous studies implemented to achieve this goal were devoted to the synthesis and modification of electrode active materials and electrolytes.⁴⁻⁶ Along with these material-based works, the design of high-mass-loading electrodes has recently garnered considerable attention as a facile and scalable architectural strategy.^{7,8}

A formidable challenge facing high-mass-loading electrodes is the acquisition of uniform ion/electron conduction pathways throughout their through-thickness direction without structural disruption.⁸⁻¹² One notable result was the anisotropic-structured porous electrodes that provided short tortuous migration paths of the electrolytes. These electrodes were fabricated by a subtractive method based on aligned sacrificial templates (magnetized nylon rods/emulsion droplets¹³ and NaCl salts¹⁴) or an additive method (three-dimensional (3D) printing,¹⁵⁻¹⁷ ice templating,^{18,19} and wood templating^{20,21}). Unfortunately, the subtractive and additive methods suffered from an inevitable increase in electrode porosity. Moreover, larger amounts of electrolyte are needed to fill the electrode pores.⁸ Consequently, unwanted loss of volumetric and gravimetric energy densities was encountered in the resulting cells. As a different approach, electron-conducting frameworks were introduced in the electrodes. Infiltrating electrode slurries into 3D conductive scaffolds²²⁻²⁴ or integrating electrode materials with conductive percolation networks²⁵⁻²⁸ improved the electronic conductivity and mechanical robustness of the electrodes. However, the weights and volumes of the electron-conducting frameworks themselves were excessively large, thus decreasing the mass loadings of electrode active materials in the electrodes.²⁹ Additionally, the complicated and high-cost manufacturing processes hindered the practical and scalable production of high-mass-loading electrodes.”

2. How is the cost of the developed electrodes and devices? In real applications, cost is as important as performance, and sometime more sensitive than performance toward commercialization. Although the referee is aware that this is a fundamental study that focused more on the scientific aspects, but I would still suggest the authors to discuss on this point given that some readers may be interested to know its cost.

→ Our deep appreciation is devoted to the reviewer’s insightful comment. As the reviewer is

well aware of, quantitative cost estimation of the developed electrodes is very challenging because this work is in the early research stage. In response to the reviewer’s comment, we prepared the cost information (based on the lab scale) of the major materials and manufacturing devices used herein.

Table. Cost information (based on the lab scale) of the major materials and manufacturing devices used in this study.

Materials/Equipment Devices	Cost
ETPTA	~ \$0.615 g ⁻¹
PEI	~ \$0.3 g ⁻¹
Electrospinning machine	~ \$10,000/ea
UV curing device	~ \$2,000/ea

Please note that the BNQS electrode does not include a polyvinylidene fluoride (PVdF) binder that is commonly used in commercial LIB electrodes. Instead, the UV-crosslinked ethoxylated trimethylolpropane tri-acrylate (ETPTA) was introduced in the BNQS electrode to act as a kind of binder. Please see the cost comparison between the PVdF and ETPTA: \$9.76 g⁻¹ for PVdF (a lab-scale grade) and \$0.615 g⁻¹ for ETPTA (a lab-scale grade), exhibiting the cost competitiveness of the ETPTA over the PVdF. Another salient feature of the BNQS electrode proposed in this study is the facile and simple fabrication process based on the removal of time-/energy-consuming electrode drying steps. In response to the reviewer’s comment, additional discussion on the cost competitiveness was provided in the revised manuscript.

[Revised manuscript]

“As a proof of concept, we choose commercially appealing high-capacity Ni-rich layered transition metal oxide (LiNi_{0.8}Co_{0.1}Mn_{0.1}O₂, NCM811) particles as a model electrode active material.^{30,31} The UV-curable gel electrolyte precursors are mixed with NCM811 particles and conductive additives without using any typical processing solvents, such as N-methyl pyrrolidone (NMP), thus enabling the removal of solvent-drying steps **in the electrode manufacturing** and a shorter time for cell aging. **Such a facile and simple fabrication process could benefit the BNQS electrode in terms of cost reduction, because the solvent-drying steps and cell aging require substantial amount of time and energy, posing a heavy burden on commercial battery makers.**”

“The paste of BNQS electrodes was composed of NCM811(LiNi_{0.8}Co_{0.1}Mn_{0.1}O₂)/carbon black/gel electrolyte precursor (= 75/5/20 (w/w/w)) without any processing solvents. The gel electrolyte precursor was prepared by mixing UV-curable ethoxylated trimethylolpropane triacrylate (ETPTA) monomer (Aldrich) (including 1 wt.% 2-hydroxy-2-methyl-1-phenyl-1-propanone (HMPP) as a photo-initiator) and high boiling point electrolyte (1M LiPF₆ in ethylene carbonate(EC)/propylene carbonate(PC) 1/1 (v/v)) at a composition ratio of 15/85 (w/w). To fabricate the BNQS electrode, the electrode paste was stencil-printed directly onto the above-prepared electroconductive-mat, in which the thickness of electrode active layer was adjusted by the thickness of a master mold. Then, another electroconductive-mat was placed on top of the printed electrode and followed by UV irradiation (Hg UV-lamp (Lichtzen) with an irradiation peak intensity of 1260 mW cm⁻²) for solidification, yielding a BNQS unit electrode. This procedure was repeatedly conducted to reach an initially designed areal-mass-loading value. After pressing to ensure intimate interfacial contact between the electrode components, a metallic-foil current-collector-free, electroconductive-mat/gel-electrolyte-embedded BNQS electrode was obtained. As a control sample for the comparative analysis, a slurry-cast electrode was fabricated by casting a slurry mixture (NCM811/polyvinylidene fluoride (PVdF)/carbon black = 92/4/4 (w/w/w) in NMP) on an Al foil. The casted electrode slurry was vacuum-dried at 120°C for 12 h and followed by pressing in order to equally match its thickness and areal-mass-loading with those of the BNQS electrode. Meanwhile, the ETPTA (acting as a kind of binder) in the BNQS electrode has cost competitiveness compared to the commercial PVdF binder (\$0.615 g⁻¹ for ETPTA (a lab-scale grade) vs. \$9.76 g⁻¹ for PVdF (a lab-scale grade)).”

3. For cell-level energy density calculation, why packaging materials were not included. As the referee know, a battery needs to be sealed as a cell, either coin cell, pouch-type cell, or other types of packaging.

→ Many thanks for the reviewer’s constructive comment. As the reviewer pointed out, a battery needs packaging substances. However, please note that many previous works on high-mass-

loading electrodes calculated the energy densities of cells without including the packaging substances (as shown in **Supplementary Table 3** of the revised manuscript). Please captured images (for the high-mass-loading electrodes and also Li-metal batteries) below.

Supplementary Note 1.

As shown in Fig. 5e of the main text, all data is in close agreement with the dashed lines plotted from the equation relating E_{SP} to C/A . This equation could be derived according to,

$$E_{SP} = \frac{E}{M_{Total_cell}} = \frac{E/A}{M_{Total_cell}/A} = \frac{E/A}{\frac{M_{Cathode}}{A} + \frac{M_{Anode}}{A} + \frac{M_{Inactive}}{A}} \quad [\text{Eq.1}]$$

where $M_{Cathode}$, M_{Anode} , and $M_{Inactive}$ are mass for cathode, anode and inactive components (Al/Cu foils, separator and electrolyte filled in cathode/anode and separator pores), respectively.

Here, E, the cell energy, can be described as follows,

$$E = \int V(t) \cdot Idt \approx V \cdot C_{cell}$$

[Eq.2]

where V and C_{cell} are the average operating voltage of the cell and the cell capacity, respectively.

Figure. A captured image from a research article published in *Nat. Energy*, **4**, 560, (2019), “High areal capacity battery electrodes enabled by segregated nanotube networks”. The energy densities (here, E_{sp}) of this work were calculated without including the coin cell packaging substances.

Supplementary Table 1 | Cell parameters.

	Anode-Free		Lithium-ion	
	Positive	Negative	Positive	Negative
Material	NMC*	-	NMC*	AG+
Thickness (μm)	46.25	-	46.25	65.5
Loading (mg/cm^2)	15.7	-	15.7	9.94
Current Collector (CC)	Aluminum	Copper	Aluminum	Copper
CC Thickness (μm)	13	8	13	8
	Cell		Cell	
Stack Mass (mg/cm^2)	43.82		64.31	
Stack Thickness (μm)	167		272.5	
Average Voltage	3.9		3.7	
Stack Energy (Wh)	0.89		0.85	
Stack Volumetric Energy Density (Wh/L)	1231		720	
Stack Specific Energy Density (Wh/kg)	469		305	

Figure. A captured image from a research article published in *Nat. Energy*, **5**, 693, (2020), “Diagnosing and correcting anode-free cell failure via electrolyte and morphological analysis”.

The stack energy densities of this work were calculated without including the pouch packaging substances.

NCM811/Li (68%) (as shown in Figure 6a). Besides, to evaluate the real energy density of our AF-LMBs, we assembled the hand-made level multi-layer anode-free pouch cells (300 mAh) where $\text{Li}_{1.31}\text{NCM811}$ (lithiated by n-butyl Li, 31.25 mg cm^{-2} , $30 \times 40 \text{ mm}^2$) paired with Cu foil ($31 \times 41 \text{ mm}^2$) (Figure 6b–d). As shown in Figure 6c, $\text{Li}_{1.31}\text{NCM811/Cu}$ outperforms NCM811/Li in initial energy density (447 Wh kg^{-1} vs. 415 Wh kg^{-1}), and the energy density remains 416 Wh kg^{-1} after 50 cycles. The energy density of this $\text{Li}_{1.31}\text{NCM811/Cu}$ cell is determined from the weight of the cell core, excluding the weight of Al-plastic film packaging (21.8 wt%).^[23] Notably, the energy density based on the total

Figure. A captured image from a research article published in *Angew Chem. Int. Ed.*, **60**, 8289, (2021), “Li-rich $\text{Li}_2[\text{Ni}_{0.8}\text{Co}_{0.1}\text{Mn}_{0.1}]\text{O}_2$ for anode-free lithium metal batteries”. The energy densities of this work were calculated without including the pouch packaging substances.

Table S3. Coin cell parameters for the energy density calculations of LMBs

Parts	Materials	Basic parameters	Total weight
Cathode	NMC811 cathode	4.2 mAh cm^{-2} areal loading, 1.27 cm^2 total area, 96 wt% active material, 3 g cm^{-3} press density, 35% porosity, $12 \text{ }\mu\text{m}$ Al foil, coating thickness $\sim 72 \text{ }\mu\text{m}$	30.58 mg
Anode	Li metal on Cu foil	$50 \text{ }\mu\text{m}$ Li metal on $8 \text{ }\mu\text{m}$ Cu foil, 1.27 cm^2 total area	13.56 mg
Separator	Polyethylene separator	1.56 mg cm^{-2} areal density, 2.11 cm^2 total area, 40% porosity	3.29 mg
Electrolyte	LHCE in this work	3 g (Ah)^{-1} E/C ratio	16.0 mg

For the $\text{Li}||\text{NMC811}$ coin cell with $50 \text{ }\mu\text{m}$ Li anode, the total weight is 63.43 mg (excluding the coin cell parts). The theoretical output voltage of the NMC811 cathode is 3.85 V, the calculated energy density of the cell is 325 Wh kg^{-1} .

For the Li-free coin cell, the total weight is 49.87 mg (excluding the coin cell parts). Therefore, the calculated energy density of the cell is 412 Wh kg^{-1} .

Figure. A captured image from a research article published in *Joule*, **3**, 1662, (2019), “Enabling High-Voltage Lithium-Metal Batteries under Practical Conditions”. The energy densities of this work were calculated without including the coin cell packaging substances.

For this reason, herein, the energy densities of the BNQS electrodes were calculated without including the packaging substances, in order to make a fair comparison with those of previously

reported high-mass-loading electrodes that excluded the packaging substances. The comparison results were shown in Fig. 5c and Supplementary Table 3 of the revised manuscript.

In response to the reviewer's comment, we estimated the energy densities (including the packaging substances) and provided the energy density values in terms of energy density (*with packaging*) and energy density (*without packaging*), respectively. These results were added in the revised manuscript (Supplementary Table 5).

[Revised manuscript]

Fig. 5 Enabling high-energy-density Li metal cells by the BNQS cathodes. **a** Charge/discharge profiles of the cells (BNQS cathode|separator|Li metal anode (100 μm corresponding to an areal capacity of 20 mAh cm^{-2})) as a function of areal-mass-loading of the BNQS cathodes at a voltage range of 3.0 – 4.4 V and charge/discharge current density of 0.05 C/0.1 C. **b** Realization of theoretical specific capacities of cathode materials as a function of areal capacity (BNQS cathodes versus previously reported cathodes). **c** Gravimetric energy

densities of cells as a function of areal capacity (BNQS cathodes versus previously reported cathodes), in which the cell weight was estimated by considering those of cathodes, anodes, separators, and electrolytes. **d Cycling performance of the pouch-type cell ($26 \times 24 \text{ mm}^2$ in size). The cell consisted of BNQS cathode (areal capacity of 12.1 mAh cm^{-2})|separator|Li metal anode (areal capacity of 20 mAh cm^{-2}), in which the N/P ratio was 1.6 and the amount of electrolyte in the cell (including the gel electrolyte in the BNQS cathode) was set as 2.3 g Ah^{-1} . The energy density (with packaging) of the pouch-type cell was estimated by considering its entire cell weight. The cell was reassembled by replacing the cycled Li metal anode with a fresh one after 70 cycles.**

“The aforementioned efficiency of capacity utilization in the electrodes crucially affects the energy densities of cells. The **energy densities (without packaging)** of the BNQS cathode-containing cells ($404 \text{ Wh kg}^{-1}/1025 \text{ Wh L}^{-1}$) were compared with those of previously reported high-mass-loading cathode systems (**Fig. 5c, Supplementary Fig. 22 and Supplementary Table 3**), in which the **energy densities were estimated without including packaging substances** (see **Supplementary Table 4** for calculation details). **The highest energy densities (without packaging) achieved by the BNQS cathode with areal capacity of 12.3 mAh cm^{-2}** exceeded those of previous studies, demonstrating the viability of the BNQS cathode approach in enabling the high-energy-density cell.

Furthermore, a pouch-type cell ($26 \times 24 \text{ mm}^2$ in size) with the BNQS cathode was fabricated **and its cell performance was investigated at a voltage range of $3.0 - 4.4 \text{ V}$** to explore the feasibility of this high-mass-loading electrode concept. **The pouch-type cell was composed of the BNQS cathode (areal capacity of 12.1 mAh cm^{-2})|separator|Li metal anode (areal capacity of 20 mAh cm^{-2}), in which the N/P ratio was 1.6. The amount of electrolyte in the cell (including the gel electrolyte in the BNQS cathode) was limited as 2.3 g Ah^{-1} to practically evaluate the cell.^{44,54}** The obtained pouch-type cell showed **an initial discharge capacity of 203 mAh g^{-1} (Supplementary Fig. 23)**, exhibiting almost full utilization of the theoretical specific capacity (205 mAh g^{-1})^{30,31} of NCM811 in the BNQS electrode. In addition, **the energy densities (with packaging) of the pouch-type cell were estimated to reach $321 \text{ Wh kg}^{-1}/772 \text{ Wh L}^{-1}$ (Fig. 5d and Supplementary Table 5)**. The pouch-type cell showed **stable cycling performance (Fig. 5d and Supplementary Fig. 23)**, underscoring the electrochemical viability of the high-mass-loading BNQS cathodes. **Meanwhile, no significant difference in the**

electrode thickness was observed before and after the cycling test (Supplementary Fig. 24), indicating the structural stability of the BNQS cathode with cycling. Intriguingly, the areal capacity of the pouch-type cell was almost recovered to an initial value, after the cycled Li metal anode was replaced with a fresh one after 70 cycles by referring to the procedure described in a previous work⁵⁵ (Fig. 5d). This result may indicate that the BNQS cathode is not a crucial cause of the cycling fading of the pouch-type cell.”

“The pouch-type Li metal cell (BNQS cathode (12.1 mAh cm⁻²)/Li metal anode (20.0 mAh cm⁻²) corresponding to an N/P ratio of 1.6) exhibited high energy densities (321 Wh kg⁻¹/772 Wh L⁻¹, including packaging substances) that lie beyond those of previously reported high-mass-loading electrodes. It is envisioned that the BNQS electrode strategy described herein will open a new route toward practical high-mass-loading electrodes with redox homogeneity in the through-thickness direction and holds great promise as a new electrode design that can be combined with next-generation electrode active materials that are struggling with conventional electrode architectures.”

“Calculation of energy densities

The energy densities (without packaging) of the cells were estimated based on the weight and volume of cathode (including electroconductive-mat interlayers and gel electrolyte), anode (including a current collector), separator, and electrolyte in the cell. Meanwhile, the energy density (with packaging) of the pouch-type cell was estimated based on the experimentally measured weight and volume of the cell (including the pouch packaging film). Calculation details for the energy densities (with and without packaging) are described in Supplementary Table 4 and 5.”

Supplementary Table 5. Parameters of the pouch-type cell, in which the cell was composed of the BNQS cathode (BNQS cathode (areal capacity of 12.1 mAh cm⁻²)|separator|Li metal anode (areal capacity of 20 mAh cm⁻²). The area of the BNQS cathode, Li metal anode and separator were 20 × 20 mm², 21 × 21 mm² and 23 × 23 mm², respectively. The cell energy densities were estimated based on the experimentally measured weight and volume of the cell

(including packaging substances).

Cell components	Mass (mg)	Areal mass (mg cm ⁻²)	Areal capacity (mAh cm ⁻²)	Thickness (μm)
BNQS cathode	330	83	12	314
Li-metal anode	64	13	20	109
Separator	7	1	-	16
Injected electrolyte	59	-	-	-
Pouch substances	106	-	-	150

Pouch-type cell parameters

Mass (mg)	566
Thickness (μm)	589
Capacity (mAh)	48
Energy (mWh)	182
Energy density (Wh kg ⁻¹)	321
Energy density (Wh L ⁻¹)	772

4. Regarding the electrode thickness and mass loading, some previous reported thick electrodes can reach up to millimeter and more than 80 mg cm⁻². Is the current design able to reach similar values? And what are the limits for the thickness and mass loading using this design?

→ Many thanks for reviewer’s valuable comments. A recent study (ref.: *Adv. Mater.* **33**, 2101275 (2021)) on the high-mass-loading electrodes theoretically calculated that the gravimetric/volumetric energy densities reach a plateau with increasing the areal-mass-loading of cathodes and eventually there exists an optimal loading for practical Li-NMC cells.

Figure 2. Key parameters dominating gravimetric energy density of a,b) Li-NMC622 and c,d) Li-S pouch cells. a) Dependence of gravimetric energy density on NMC622 loading and NMC622 fraction (cathode porosity = 40%) and b) dependence of gravimetric energy density on NMC622 loading and cathode porosity (NMC622 fraction = 90 wt%). c) Dependence of gravimetric energy density on S loading and S fraction (cathode porosity = 50%) and d) dependence of gravimetric energy density on S loading and cathode porosity (S fraction = 60 wt%). See Table S1 (Supporting Information) for more details on assumptions made in calculating performance metrics.

porosities. Interestingly, the cathode porosity seems to have a bigger impact on the volumetric energy density compared to the gravimetric energy density. For instance, at 30 mg cm⁻² loading (Figure 2a, dash lines). It means that there exists an optimal loading for practical Li-NMC pouch cells. The optimal

Figure. A captured image from a research article published in *Adv. Mater.* **33**, 2101275 (2021), “From fundamental understanding to engineering design of high-performance thick electrodes for scalable energy-storage systems”.

Similar to the results of the previous studies mentioned above, the energy densities of the BNQS electrode tend to be gradually saturated with increasing the areal capacity (as shown in **Fig. 5c**), revealing that further increase of areal capacity over a critical point may not effectively contribute to energy density. In response to the reviewer’s comment, we further increased areal-mass-loading values of BNQS electrodes using the same materials and

fabrication technique. A BNQS electrode with an areal-mass-loading of 90 mg cm^{-2} was successfully obtained, as shown below. This indicates that the thickness and mass loading of electrodes can be further increased through the BNQS electrode design without serious difficulty. Meanwhile, as described in **Fig. 5b**, realizing the theoretical capacities of electrode materials is a prerequisite to enabling the high-energy-density cells. Please note that the BNQS electrode with an areal-mass-loading of 90 mg cm^{-2} showed a specific capacity of $185 \text{ mAh g}_{\text{NCM811}}^{-1}$, which is slightly lower than the theoretical capacity ($205 \text{ mAh g}_{\text{NCM811}}^{-1}$, ref. [30, 31]) of NCM811 active materials. As a result, an energy density (without packaging) of the BNQS electrode with an areal-mass-loading of 90 mg cm^{-2} led to $405 \text{ Wh kg}_{\text{cell}}^{-1}$, disclosing that there exists an optimum areal-mass-loading in the BNQS electrode in terms of realizing theoretical specific capacity of electrode active materials. This result suggests that the areal-mass-loading of electrodes should be rationally designed based on the consideration of energy density of cells.

Figure. a Photograph of the BNQS electrode with an areal-mass-loading of 90 mg cm^{-2} . **b** Charge/discharge profiles of the BNQS electrode at a current density of $0.05 \text{ C}/0.1 \text{ C}$ ($= 1.85 \text{ mA cm}^{-2}$).

5. A mass loading of 36 mg cm^{-2} is not the highest for slurry-casted electrode. Some previous works reported higher values. So please be caution in making such claim.

→ Our deep appreciation is devoted to the reviewer's valuable comment. Please accept my apology for confusing the reviewer on this matter. The areal-mass-loading of 36 mg cm^{-2} was the highest value obtained with the slurry-cast electrodes of this study (not including the areal-mass-loadings reported in previous works). In response to the reviewer's comment, the sentence was revised.

[Revised manuscript]

“The 3D microstructures of the BNQS and slurry-cast electrodes were investigated using focused-ion-beam nanotomography and associated 3D reconstruction techniques. For a fair comparison between the two electrodes, the electrode thickness was set to $160 \text{ }\mu\text{m}$ (areal-mass loading of 36 mg cm^{-2}), **in which the slurry-cast electrode as well as the BNQS electrode maintained its structural integrity.**”

“Fig. 2 Structural analysis of the BNQS electrode (versus slurry-cast electrode). 3D microstructural analysis, focusing on carbon (colored in yellow) distribution, of a slurry-cast and b BNQS electrodes, in which the electrode thickness was set as $160 \text{ }\mu\text{m}$ (areal-mass-loading of 36 mg cm^{-2}), to make a fair comparison between the two electrodes. c Volume fraction of carbon in each section of the electrode along its z -axis direction. **d** Electronic conductivity of the electrodes. **e** Nyquist plots obtained by a symmetric cell configuration at 0% SOC. Cross-sectional SEM images and corresponding localized charge transfer resistance (obtained by LEIS analysis) of **f** slurry-cast and **g** BNQS electrodes.”

6. The controlled slurry-casted electrode should also be pressed to make better contact of electrode materials and reduce the porosity for fairer comparison, as the layer-by-layer structured electrodes were pressed for these purposes.

→ Many thanks for the reviewer's valuable comment. As described in the manuscript, to make a fair comparison between the BNQS and slurry-cast electrodes, the electrode thickness was set to $160 \text{ }\mu\text{m}$ under the same areal-mass loading (36 mg cm^{-2}). To reach this condition, the

BNQS and slurry-cast electrodes were subjected to the pressing. In response to the reviewer's comment, additional sentence describing the pressing of the control slurry-cast electrodes was provided in the revised manuscript.

[Revised manuscript]

“On top of the electroconductive mat that can act as an alternative porous current collector replacing the conventional metallic-foil one, the above-prepared electrode paste was stencil-printed, followed by UV curing, producing a BNQS unit electrode (areal-mass (solely including NCM811, if not specified) loading $\sim 10 \text{ mg cm}^{-2}$). The optimal thicknesses (40 \sim 60 μm) of active layers in the unit electrodes were determined by analyzing the electronic conductivity of the unit electrodes as a function of thickness of active layer (**Supplementary Fig. 4**). Subsequently, the stencil-printing-based electrode fabrication step was repeated, together with insertion of the electroconductive mats in the through-thickness direction, until reaching the mass-loading value of interest (**Fig. 1a**). After pressing to ensure intimate interfacial contact between the electrode components, a metallic-foil current-collector-free, electroconductive-mat/gel-electrolyte-embedded BNQS electrode was obtained (**Supplementary Fig. 5**). **The basic information of the electroconductive-mat interlayers inserted in the BNQS electrodes was provided as a function of areal-mass-loading (Supplementary Table 1).** We note that the BNQS electrode already includes 20 wt.% gel electrolyte (comprising the UV-cured ETPTA polymer and 1M LiPF₆ in EC/PC) besides NCM811 and carbon black additives, which could correspond to binders and liquid electrolytes embedded in pores of conventional LIB electrodes. The contents of binders and liquid electrolytes in conventional LIB electrodes are generally in the range of 15 – 20 wt.%,³⁶ which are not significantly different from the gel electrolyte content of the BNQS electrode. Excluding the liquid electrolyte from the gel electrolyte leads to an electrode composition of NCM811/carbon black/ETPTA (acting as a kind of binder) = 90.5/6/3.5 (w/w/w), revealing that the NCM811 content in the BNQS electrode appears comparable to those of previously reported electrodes. Meanwhile, a control electrode was fabricated by casting a slurry (NCM811/carbon black additive/polyvinylidene fluoride (PVdF) binder = 92/4/4 (w/w/w) in NMP as a processing solvent) on an aluminum (Al) current collector **and followed by roll-pressing to equally match its thickness and areal-mass-loading with those of the BNQS electrode**. In comparison with the BNQS electrode, the slurry-cast electrode failed to reach a thickness of higher than 180 μm because of unwanted crack formation and delamination during

the drying of NMP (Fig. 1b).^{9,10}”

“Fabrication of BNQS electrodes

The electroconductive-mat interlayer was fabricated by concurrent electrospinning (for polyetherimide-trimethylolpropane propoxylate triacrylate (PEI-TPPTA) nanofibers) and electrospaying (for SWCNT, TUBALLTM) through two different nozzles. For the electrospinning, PEI powder/TPPTA monomer (= 70/30 (w/w)) were dissolved in N-methyl-2-pyrrolidone (NMP, Aldrich)/dimethylacetamide (DMAc) (= 25/75 (w/w)) with benzoyl peroxide (BPO) as a thermal-initiator. For the electrospaying, the SWCNT suspension (solid content = 1 wt.% in water/isopropyl alcohol (IPA) (=90/10 (w/w)) was used, in which 1 wt.% of polyvinylpyrrolidone (PVP) was incorporated as a dispersion agent. The detailed processing conditions were 11 keV with a feed rate of 3 $\mu\text{L min}^{-1}$ for the electrospinning and 14 keV with a feed rate of 60 $\mu\text{L min}^{-1}$ for the electrospaying. The obtained mat was thermally cured at 90°C for 3 h and then roll-pressed at 80°C, producing a self-standing electroconductive interlayer.

The paste of BNQS electrodes was composed of NCM811(LiNi_{0.8}Co_{0.1}Mn_{0.1}O₂)/carbon black/gel electrolyte precursor (= 75/5/20 (w/w/w)) without any processing solvents. The gel electrolyte precursor was prepared by mixing UV-curable ethoxylated trimethylolpropane triacrylate (ETPTA) monomer (Aldrich) (including 1 wt.% 2-hydroxy-2-methyl-1-phenyl-1-propanone (HMPP) as a photo-initiator) and high boiling point electrolyte (1M LiPF₆ in ethylene carbonate(EC)/propylene carbonate(PC) 1/1 (v/v)) at a composition ratio of 15/85 (w/w). To fabricate the BNQS electrode, the electrode paste was stencil-printed directly onto the above-prepared electroconductive-mat, in which the thickness of electrode active layer was adjusted by the thickness of a master mold. Then, another electroconductive-mat was placed on top of the printed electrode and followed by UV irradiation (Hg UV-lamp (Lichtzen) with an irradiation peak intensity of 1260 mW cm^{-2}) for solidification, yielding a BNQS unit electrode. This procedure was repeatedly conducted to reach an initially designed areal-mass-loading value. **After pressing to ensure intimate interfacial contact between the electrode components, a metallic-foil current-collector-free, electroconductive-mat/gel-electrolyte-embedded BNQS electrode was obtained.** As a control sample for the comparative analysis, a slurry-cast electrode was fabricated by casting a slurry mixture (NCM811/polyvinylidene fluoride (PVdF)/carbon black = 92/4/4 (w/w/w) in NMP) on an Al foil. The casted electrode

slurry was vacuum-dried at 120°C for 12 h and followed by pressing **in order to equally match its thickness and areal-mass-loading with those of the BNQS electrode.**”

Reviewer #2

After carefully reading this manuscript, this reviewer found that there are some critical problems. So, this work is not suitable for publication.

1. Most of results are based on the coin cell (only one charge-discharge curve in figure 5D is based on pouch-type cell). The practice energy density of the cell is calculated on the cathode, anode, electrolyte and separator excluding weights of package of coin cell and Al/Cu foil films, which is not acceptable.

→ Our deep appreciation is devoted to the reviewer's valuable comments. Many previous works calculated the energy densities without including the packaging substances (as shown in **Supplementary Table 3** of the revised manuscript). Please see the captured image below. For this reason, herein, the energy densities of the BNQS electrodes were calculated without including the packaging substances, in order to make a fair comparison with those of previously reported high-mass-loading electrodes that excluded the packaging substances. The comparison results were shown in **Fig. 5c** and **Supplementary Table 3** of the revised manuscript.

Supplementary Note 1.

As shown in Fig. 5e of the main text, all data is in close agreement with the dashed lines plotted from the equation relating E_{SP} to C/A . This equation could be derived according to,

$$E_{SP} = \frac{E}{M_{Total_cell}} = \frac{E/A}{M_{Total_cell}/A} = \frac{E/A}{\frac{M_{Cathode}}{A} + \frac{M_{Anode}}{A} + \frac{M_{Inactive}}{A}} \quad [Eq.1]$$

where $M_{Cathode}$, M_{Anode} , and $M_{Inactive}$ are mass for cathode, anode and inactive components (Al/Cu foils, separator and electrolyte filled in cathode/anode and separator pores), respectively.

Here, E , the cell energy, can be described as follows,

$$E = \int V(t) \cdot Idt \approx V \cdot C_{cell}$$

[Eq.2]

where V and C_{cell} are the average operating voltage of the cell and the cell capacity, respectively.

Figure. A captured image from a research article published in *Nat. Energy*, **4**, 560, (2019), “High areal capacity battery electrodes enabled by segregated nanotube networks”. The energy densities (here, E_{sp}) of this work were calculated without including the packaging substances.

Supplementary Table 1 | Cell parameters.

	Anode-Free		Lithium-ion	
	Positive	Negative	Positive	Negative
Material	NMC*	-	NMC*	AG+
Thickness (μm)	46.25	-	46.25	65.5
Loading (mg/cm^2)	15.7	-	15.7	9.94
Current Collector (CC)	Aluminum	Copper	Aluminum	Copper
CC Thickness (μm)	13	8	13	8
	Cell		Cell	
Stack Mass (mg/cm^2)	43.82		64.31	
Stack Thickness (μm)	167		272.5	
Average Voltage	3.9		3.7	
Stack Energy (Wh)	0.89		0.85	
Stack Volumetric	1231		720	
Energy Density (Wh/L)				
Stack Specific Energy	469		305	
Density (Wh/kg)				

Figure. A captured image from a research article published in *Nat. Energy*, **5**, 693, (2020), “Diagnosing and correcting anode-free cell failure via electrolyte and morphological analysis”. The stack energy densities of this work were calculated without including the pouch packaging substances.

NCM811/Li (68%) (as shown in Figure 6a). Besides, to evaluate the real energy density of our AF-LMBs, we assembled the hand-made level multi-layer anode-free pouch cells (300 mAh) where $\text{Li}_{1.31}\text{NCM811}$ (lithiated by n-butyl Li, 31.25 mg cm^{-2} , $30 \times 40 \text{ mm}^2$) paired with Cu foil ($31 \times 41 \text{ mm}^2$) (Figure 6b–d). As shown in Figure 6c, $\text{Li}_{1.31}\text{NCM811}/\text{Cu}$ outperforms NCM811/Li in initial energy density (447 Wh kg^{-1} vs. 415 Wh kg^{-1}), and the energy density remains 416 Wh kg^{-1} after 50 cycles. The energy density of this $\text{Li}_{1.31}\text{NCM811}/\text{Cu}$ cell is determined from the weight of the cell core, excluding the weight of Al-plastic film packaging (21.8 wt%).^[23] Notably, the energy density based on the total

Figure. A captured image from a research article published in *Angew Chem. Int. Ed.*, **60**, 8289, (2021), “Li-rich $\text{Li}_2[\text{Ni}_{0.8}\text{Co}_{0.1}\text{Mn}_{0.1}]\text{O}_2$ for anode-free lithium metal batteries”. The energy densities of this work were calculated without including the pouch packaging substances.

Table S3. Coin cell parameters for the energy density calculations of LMBs

Parts	Materials	Basic parameters	Total weight
Cathode	NMC811 cathode	4.2 mAh cm ⁻² areal loading, 1.27 cm ² total area, 96 wt% active material, 3 g cm ⁻³ press density, 35% porosity, 12 μm Al foil, coating thickness ~72 μm	30.58 mg
Anode	Li metal on Cu foil	50 μm Li metal on 8 μm Cu foil, 1.27 cm ² total area	13.56 mg
Separator	Polyethylene separator	1.56 mg cm ⁻² areal density, 2.11 cm ² total area, 40% porosity	3.29 mg
Electrolyte	LHCE in this work	3 g (Ah) ⁻¹ E/C ratio	16.0 mg

For the Li||NMC811 coin cell with 50 μm Li anode, the total weight is 63.43 mg (excluding the coin cell parts). The theoretical output voltage of the NMC811 cathode is 3.85 V, the calculated energy density of the cell is 325 Wh kg⁻¹.

For the Li-free coin cell, the total weight is 49.87 mg (excluding the coin cell parts). Therefore, the calculated energy density of the cell is 412 Wh kg⁻¹.

Figure. A captured image from a research article published in *Joule*, **3**, 1662, (2019), “Enabling High-Voltage Lithium-Metal Batteries under Practical Conditions”. The energy densities of this work were calculated without including the coin cell packaging substances.

Different from the reviewer’s comment, the energy densities presented in this study included the weight of current collectors, which were already described in the original manuscript. To avoid any misunderstanding, additional description was provided in **Supplementary Table 4**. In response to the reviewer’s comment, we estimated the energy densities of pouch-type cells and provided their energy density values based on the experimentally measured weight and volume of the cell (with packaging). Notably, the energy density (with packaging) of the pouch-type cell was estimated (**Supplementary Table 5**) and added in the revised manuscript.

[Revised manuscript]

“The aforementioned efficiency of capacity utilization in the electrodes crucially affects the energy densities of cells. The **energy densities (without packaging)** of the BNQS cathode-containing cells (**404 Wh kg⁻¹/1025 Wh L⁻¹**) were compared with those of previously reported high-mass-loading cathode systems (**Fig. 5c, Supplementary Fig. 22 and Supplementary Table 3**), in which the **energy densities were estimated without including packaging substances** (see **Supplementary Table 4** for calculation details). **The highest energy densities (without**

packaging) achieved by the BNQS cathode with areal capacity of 12.3 mAh cm^{-2} exceeded those of previous studies, demonstrating the viability of the BNQS cathode approach in enabling the high-energy-density cell.

Furthermore, a pouch-type cell ($26 \times 24 \text{ mm}^2$ in size) with the BNQS cathode was fabricated and its cell performance was investigated at a voltage range of $3.0 - 4.4 \text{ V}$ to explore the feasibility of this high-mass-loading electrode concept. The pouch-type cell was composed of the BNQS cathode (areal capacity of 12.1 mAh cm^{-2})|separator|Li metal anode (areal capacity of 20 mAh cm^{-2}), in which the N/P ratio was 1.6. The amount of electrolyte in the cell (including the gel electrolyte in the BNQS cathode) was limited as 2.3 g Ah^{-1} to practically evaluate the cell.^{44,54} The obtained pouch-type cell showed an initial discharge capacity of 203 mAh g^{-1} (Supplementary Fig. 23), exhibiting almost full utilization of the theoretical specific capacity (205 mAh g^{-1})^{30,31} of NCM811 in the BNQS electrode. In addition, the energy densities (with packaging) of the pouch-type cell were estimated to reach $321 \text{ Wh kg}^{-1}/772 \text{ Wh L}^{-1}$ (Fig. 5d and Supplementary Table 5). The pouch-type cell showed stable cycling performance (Fig. 5d and Supplementary Fig. 23), underscoring the electrochemical viability of the high-mass-loading BNQS cathodes. Meanwhile, no significant difference in the electrode thickness was observed before and after the cycling test (Supplementary Fig. 24), indicating the structural stability of the BNQS cathode with cycling. Intriguingly, the areal capacity of the pouch-type cell was almost recovered to an initial value, after the cycled Li metal anode was replaced with a fresh one after 70 cycles by referring to the procedure described in a previous work⁵⁵ (Fig. 5d). This result may indicate that the BNQS cathode is not a crucial cause of the cycling fading of the pouch-type cell.”

“Calculation of energy densities

The energy densities (without packaging) of the cells were estimated based on the weight and volume of cathode (including electroconductive-mat interlayers and gel electrolyte), anode (including a current collector), separator, and electrolyte in the cell. Meanwhile, the energy density (with packaging) of the pouch-type cell was estimated based on the experimentally measured weight and volume of the cell (including the pouch packaging film). Calculation details for the energy densities (with and without packaging) are described in Supplementary Table 4 and 5.”

Supplementary Table 4. Calculation details for the gravimetric/volumetric energy densities of cells containing the BNQS cathodes.

As shown in **Fig. 5c**, the gravimetric energy density of the Li metal cell is plotted. The equation²⁵ be derived according to,

[Eq.1] Gravimetric energy density (Wh kg⁻¹)

$$= \frac{\text{Energy}}{\text{Mass of cell}} = \frac{\frac{\text{Energy}}{\text{Area}}}{\frac{\text{Mass of cell}}{\text{Area}}} = \frac{\text{Nominal Volatge} \times C/A}{M_{\text{cathode}}/A + M_{\text{anode}}/A + M_{\text{separator}}/A + M_{\text{electrolyte}}/A}$$

where M_{cathode} , M_{anode} , $M_{\text{separator}}$ and $M_{\text{electrolyte}}$ are the mass of cathode (including the **electroconductive-mat interlayers** and gel electrolyte), anode (consisting of Li metal (100 μm) corresponding to an areal capacity of 20 mAh cm⁻²) and Cu current collector (9 μm)), separator and injected electrolyte. C and A indicates capacity and area, respectively.

Supplementary Table 5. Parameters of the pouch-type cell, in which the cell was composed of the BNQS cathode (BNQS cathode (areal capacity of 12.1 mAh cm⁻²)|separator|Li metal anode (areal capacity of 20 mAh cm⁻²). The area of the BNQS cathode, Li metal anode and separator were 20 × 20 mm², 21 × 21 mm² and 23 × 23 mm², respectively. The cell energy densities were estimated based on the experimentally measured weight and volume of the cell (including packaging substances).

Cell components	Mass (mg)	Areal mass (mg cm ⁻²)	Areal capacity (mAh cm ⁻²)	Thickness (μm)
BNQS cathode	330	83	12	314
Li-metal anode	64	13	20	109
Separator	7	1	-	16
Injected electrolyte	59	-	-	-
Pouch substances	106	-	-	150

Pouch-type cell parameters

Mass (mg)	566
Thickness (μm)	589
Capacity (mAh)	48
Energy (mWh)	182
Energy density (Wh kg^{-1})	321
Energy density (Wh L^{-1})	772

2. The results on the coin cell are impossible to scale up for pouch-type cell. That is why there are only one charge-discharge curve in figure 5D. It is better to show the cycle performance of pouch-type cell.

→ Thank you so much for the reviewer's valuable comment. In response to the reviewer's comment, the cycling performance of the pouch-type cell was investigated using a pouch-type cell at a voltage range of 3.0 – 4.4 V. Please see **Fig. 5d** in the revised manuscript. The pouch-type cell was composed of the BNQS cathode (areal capacity of 60 mg cm^{-2})|separator|Li metal anode (areal capacity of 20 mAh cm^{-2}), in which the N/P ratio was 1.6. The amount of electrolyte in the cell (including the gel electrolyte in the BNQS cathode) was limited as 2.3 g Ah^{-1} to practically evaluate the cell (ref.: *Nat. Energy*, **4**, 180, (2019) and *Joule*, **3**, 1094, (2019)). The resulting pouch-type cell showed a discharge capacity of 203 mAh g^{-1} , exhibiting almost full utilization of the theoretical specific capacity (205 mAh g^{-1} , ref. [30, 31]) of NCM811 in the BNQS electrode. In addition, the pouch-type cell showed stable cycling performance (**Figs. 5D** and **Supplementary Fig. 23**), underscoring the electrochemical

viability of the high-mass-loading BNQS cathodes. Meanwhile, the areal capacity of the pouch-type cell was almost recovered to an initial value, after the Li metal anode was replaced with a fresh one by referring to the procedure described in a previous work (ref.: *Joule* **3**, 1662 (2019)) (Fig. 5d). This result may indicate that the BNQS cathode is not a crucial cause of the cycling fading of the pouch-type cell.

[Revised manuscript]

Fig. 5 Enabling high-energy-density Li metal cells by the BNQS cathodes. a Charge/discharge profiles of the cells (BNQS cathode|separator|Li metal anode (100 μm corresponding to an areal capacity of 20 mAh cm^{-2})) as a function of areal-mass-loading of the BNQS cathodes at a voltage range of 3.0 – 4.4 V and charge/discharge current density of 0.05 C/0.1 C. **b** Realization of theoretical specific capacities of cathode materials as a function of areal capacity (BNQS cathodes versus previously reported cathodes). **c** Gravimetric energy densities of cells as a function of areal capacity (BNQS cathodes versus previously reported

cathodes), in which the cell weight was estimated by considering those of cathodes, anodes, separators, and electrolytes. **d** Cycling performance of the pouch-type cell ($26 \times 24 \text{ mm}^2$ in size). The cell consisted of BNQS cathode (areal capacity of 12.1 mAh cm^{-2})|separator|Li metal anode (areal capacity of 20 mAh cm^{-2}), in which the N/P ratio was 1.6 and the amount of electrolyte in the cell (including the gel electrolyte in the BNQS cathode) was set as 2.3 g Ah^{-1} . The energy density (with packaging) of the pouch-type cell was estimated by considering its entire cell weight. The cell was reassembled by replacing the cycled Li metal anode with a fresh one after 70 cycles.

Supplementary Fig. 23 Charge/discharge profiles of the pouch-type cell ($26 \times 24 \text{ mm}^2$ in size) composed of BNQS cathode (areal capacity of 12.1 mAh cm^{-2})|separator|Li metal anode (areal capacity of 20 mAh cm^{-2}) at 1st, 30th, 50th, and 70th cycle. To enable development of the practical cell, the N/P ratio was 1.6, while the E/C ratio was set as 2.3 g Ah^{-1} .

“The aforementioned efficiency of capacity utilization in the electrodes crucially affects the energy densities of cells. The **energy densities (without packaging)** of the BNQS cathode-containing cells ($404 \text{ Wh kg}^{-1}/1025 \text{ Wh L}^{-1}$) were compared with those of previously reported high-mass-loading cathode systems (Fig. 5c, Supplementary Fig. 22 and Supplementary Table 3), in which the **energy densities were estimated without including packaging substances** (see Supplementary Table 4 for calculation details). **The highest energy densities (without packaging)** achieved by the BNQS cathode with areal capacity of 12.3 mAh cm^{-2} exceeded

those of previous studies, demonstrating the viability of the BNQS cathode approach in enabling the high-energy-density cell.

Furthermore, a pouch-type cell ($26 \times 24 \text{ mm}^2$ in size) with the BNQS cathode was fabricated and its cell performance was investigated at a voltage range of 3.0 – 4.4 V to explore the feasibility of this high-mass-loading electrode concept. The pouch-type cell was composed of the BNQS cathode (areal capacity of 12.1 mAh cm^{-2})|separator|Li metal anode (areal capacity of 20 mAh cm^{-2}), in which the N/P ratio was 1.6. The amount of electrolyte in the cell (including the gel electrolyte in the BNQS cathode) was limited as 2.3 g Ah^{-1} to practically evaluate the cell.^{44,54} The obtained pouch-type cell showed an initial discharge capacity of 203 mAh g^{-1} (Supplementary Fig. 23), exhibiting almost full utilization of the theoretical specific capacity (205 mAh g^{-1})^{30,31} of NCM811 in the BNQS electrode. In addition, the energy densities (with packaging) of the pouch-type cell were estimated to reach $321 \text{ Wh kg}^{-1}/772 \text{ Wh L}^{-1}$ (Fig. 5d and Supplementary Table 5). The pouch-type cell showed stable cycling performance (Fig. 5d and Supplementary Fig. 23), underscoring the electrochemical viability of the high-mass-loading BNQS cathodes. Meanwhile, no significant difference in the electrode thickness was observed before and after the cycling test (Supplementary Fig. 24), indicating the structural stability of the BNQS cathode with cycling. Intriguingly, the areal capacity of the pouch-type cell was almost recovered to an initial value, after the cycled Li metal anode was replaced with a fresh one after 70 cycles by referring to the procedure described in a previous work⁵⁵ (Fig. 5d). This result may indicate that the BNQS cathode is not a crucial cause of the cycling fading of the pouch-type cell.”

3. The gravimetric and volumetric energy density are based on the calculation method shown in Table S3. This is not acceptable. The practice energy densities should be based on the measured weight and volume of the cell.

→ Many thanks for the reviewer’s comment. As replied in the reviewer’s comment #1, the energy densities of the coin-type cell were estimated without including the packaging substances, in order to make a fair comparison with those of previously reported high-mass-loading electrodes that excluded the packaging substances. In response to the reviewer’s

comment, the energy density of the pouch-type cell was estimated based on the experimentally measured weight and volume of the cell (definitely, including the current collectors and packaging substances). The detailed data used to estimate the energy density of the pouch-type cell were added in **Supplementary Table 5** of the revised manuscript. The reviewer's constructive comment is highly appreciated, again.

[Revised manuscript]

“The aforementioned efficiency of capacity utilization in the electrodes crucially affects the energy densities of cells. The **energy densities (without packaging)** of the BNQS cathode-containing cells (**404 Wh kg⁻¹/1025 Wh L⁻¹**) were compared with those of previously reported high-mass-loading cathode systems (**Fig. 5c, Supplementary Fig. 22 and Supplementary Table 3**), in which the **energy densities were estimated without including packaging substances** (see **Supplementary Table 4** for calculation details). **The highest energy densities (without packaging) achieved by the BNQS cathode with areal capacity of 12.3 mAh cm⁻²** exceeded those of previous studies, demonstrating the viability of the BNQS cathode approach in enabling the high-energy-density cell.

Furthermore, a pouch-type cell (26 × 24 mm² in size) with the BNQS cathode was fabricated **and its cell performance was investigated at a voltage range of 3.0 – 4.4 V** to explore the feasibility of this high-mass-loading electrode concept. **The pouch-type cell was composed of the BNQS cathode (areal capacity of 12.1 mAh cm⁻²)|separator|Li metal anode (areal capacity of 20 mAh cm⁻²), in which the N/P ratio was 1.6. The amount of electrolyte in the cell (including the gel electrolyte in the BNQS cathode) was limited as 2.3 g Ah⁻¹ to practically evaluate the cell.^{44,54} The obtained pouch-type cell showed **an initial discharge capacity of 203 mAh g⁻¹ (Supplementary Fig. 23), exhibiting almost full utilization of the theoretical specific capacity (205 mAh g⁻¹)^{30,31} of NCM811 in the BNQS electrode.** In addition, **the energy densities (with packaging) of the pouch-type cell were estimated to reach 321 Wh kg⁻¹/772 Wh L⁻¹ (Fig. 5d and Supplementary Table 5).** The pouch-type cell showed **stable cycling performance (Fig. 5d and Supplementary Fig. 23),** underscoring the electrochemical viability of the high-mass-loading BNQS cathodes. **Meanwhile, no significant difference in the electrode thickness was observed before and after the cycling test (Supplementary Fig. 24), indicating the structural stability of the BNQS cathode with cycling. Intriguingly, the areal capacity of the pouch-type cell was almost recovered to an initial value, after the cycled Li****

metal anode was replaced with a fresh one after 70 cycles by referring to the procedure described in a previous work⁵⁵ (Fig. 5d). This result may indicate that the BNQS cathode is not a crucial cause of the cycling fading of the pouch-type cell.”

“**Calculation of energy densities**

The energy densities (without packaging) of the cells were estimated based on the weight and volume of cathode (including electroconductive-mat interlayers and gel electrolyte), anode (including a current collector), separator, and electrolyte in the cell. Meanwhile, the energy density (with packaging) of the pouch-type cell was estimated based on the experimentally measured weight and volume of the cell (including the pouch packaging film). Calculation details for the energy densities (with and without packaging) are described in **Supplementary Table 4 and 5.**”

Supplementary Table 5. Parameters of the pouch-type cell, in which the cell was composed of the BNQS cathode (BNQS cathode (areal capacity of 12.1 mAh cm⁻²)|separator|Li metal anode (areal capacity of 20 mAh cm⁻²). The area of the BNQS cathode, Li metal anode and separator were 20 × 20 mm², 21 × 21 mm² and 23 × 23 mm², respectively. The cell energy densities were estimated based on the experimentally measured weight and volume of the cell (including packaging substances).

Cell components	Mass (mg)	Areal mass (mg cm ⁻²)	Areal capacity (mAh cm ⁻²)	Thickness (μm)
BNQS cathode	330	83	12	314
Li-metal anode	64	13	20	109
Separator	7	1	-	16
Injected electrolyte	59	-	-	-
Pouch substances	106	-	-	150

Pouch-type cell parameters

Mass	566
------	-----

(mg)	
Thickness	589
(μm)	
Capacity	48
(mAh)	
Energy	182
(mWh)	
Energy density	321
(Wh kg^{-1})	
Energy density	772
(Wh L^{-1})	

4. Based on the composition of NCM811(LiNi_{0.8}Co_{0.1}Mn_{0.1}O₂)/carbon black/gel 370 electrolyte precursor (=75/5/20 (w/w/w)), the energy density of pouch-type cell should be difficult to reach 410Wh/kg.

→ Many thanks for reviewer's valuable comment. The question of the composition ratio asked by the reviewer may stem from the misunderstanding. As described in the original manuscript and also the reviewer pointed out, the composition ratio of the BNQS electrode already included the gel electrolyte, which was different from composition ratios (typically expressed as electrode active materials/conductive additives/binders) of conventional electrodes. Actually, to avoid this unwanted misunderstanding, this issue was mentioned in the original manuscript and two different composition ratios of the BNQS electrode were provided. Excluding the liquid electrolyte from the gel electrolyte led to an electrode composition of NCM811/carbon black/ETPTA (acting as a kind of binder) = 90.5/6/3.5 (w/w/w), revealing that the NCM811 content in the BNQS electrode appeared comparable to those of previously reported electrodes. Please see the sentences below.

[Original manuscript]

“We note that the BNQS electrode already includes 20 wt.% gel electrolyte (comprising the UV-cured ETPTA polymer and 1M LiPF₆ in EC/PC) besides NCM811 and carbon black additives, which could correspond to binders and liquid electrolytes embedded in pores of conventional LIB electrodes. The contents of binders and liquid electrolytes in conventional LIB electrodes are generally in the range of 15 – 20 wt.%,³⁵ which are not significantly different from the gel electrolyte content of the BNQS electrode. Excluding the liquid electrolyte from the gel electrolyte leads to an electrode composition of NCM811/carbon black/ETPTA (acting as a kind of binder) = 90.5/6/3.5 (w/w/w), revealing that the NCM811 content in the BNQS electrode appears comparable to those of previously reported electrodes.”

Reviewer#3

In the manuscript entitled "Redox-homogeneous, quasi-solid-state high-mass-loading electrodes for high-energy-density lithium batteries", the authors reported NCM811 based novel cathode with high mass loading for lithium ion battery with high energy density (> 400 Wh/kg). It is very interesting to see that the optimized NCM811 cathode exhibits high conductivity, homogenous microstructure, and hence excellent redox reactions with multilayer connected by the conductive mat. Although this study shows promise for the practical application of the golden NCM811, the authors are needed to address the following major issues before going forward for publication.

1. The main novel point of this work is the capability to fabricate high mass loading (59.3 mg/cm²) cathode with high electrochemical performance. However, the major results are based on the electrode with mass loading of 36 mg cm⁻², in order to compare with the conventional slurry based cathode. To demonstrate the capability, cathode with 59.3 mg cm⁻² should be also characterized by electrochemical tests such as discharge with varied rates; Li diffusion coefficient; cycling performance, etc., as well as SEM tests.

→ Thank you so much for reviewer's valuable comment. In response to the reviewer's comment, additional electrochemical characterization of the BNQS cathode with the areal-mass-loading of 60 mg cm⁻² was conducted, which was provided in **Supplementary Fig. 19** and **20**. The cross-sectional SEM image of the BNQS cathode with the areal-mass-loading of 60 mg cm⁻² was shown in **Supplementary Fig. 19** as well as **Fig. 1A**. The Li ion diffusion coefficient (D_{Li^+}) of the BNQS cathode with the areal-mass-loading of 60 mg cm⁻² was measured to be 1.63×10^{-8} cm² s⁻¹ (**Supplementary Fig. 20**), which is comparable to that of the BNQS cathode with the areal-mass-loading of 36 mg cm⁻² (shown in **Fig. 3d**). The rate capability of the BNQS electrode with the areal-mass-loading of 60 mg cm⁻² was provided in **Supplementary Fig. 20b**. Under such high areal-mass-loading, the BNQS electrode showed a specific capacity of 190 mAh g⁻¹ at a discharge rate of 0.1 C ($= 1.16$ mA cm⁻²) and a voltage range of $3.0 - 4.2$ V, which almost reached the theoretical capacity (~ 192 mAh g⁻¹, ref. [46]) of NCM811. This result verifies almost full utilization of NCM811 active materials in the high-mass-loading (60 mg cm⁻²) cathode. The discharge capacity at a fast current density of 0.5 C was relatively low, which may be due to the high current density (0.5 C $= 5.8$ mA cm⁻²) and long tortuous pathways of ions and electrons. Future studies will be devoted to further

enhancing the rate capability via fine-tuning of electrode structure and materials chemistry. The cycling performance of the BNQS cathode with the areal-mass-loading of 60 mg cm^{-2} was examined using a pouch-type cell and shown in the newly revised **Fig. 5d**. The pouch-type cell was composed of the BNQS cathode (areal capacity of 12.1 mAh cm^{-2})|separator|Li metal anode (areal capacity of 20 mAh cm^{-2}), in which the N/P ratio was 1.6. The amount of electrolyte in the cell (including the gel electrolyte in the BNQS cathode) was limited as 2.3 g Ah^{-1} to practically evaluate the cell (ref.: *Nat. Energy*, **4**, 180, (2019) and *Joule*, **3**, 1094, (2019)). The resulting pouch-type cell showed a discharge capacity of 203 mAh g^{-1} at a voltage range of 3.0 – 4.4 V, exhibiting almost full utilization of the theoretical specific capacity (205 mAh g^{-1} , ref. [30, 31]) of NCM811 in the BNQS electrode. The pouch-type cell showed stable cycling performance (**Figs. 5d** and **Supplementary Fig. 23**). Moreover, the energy densities (with packaging) of the pouch-type cell were estimated to reach $321 \text{ Wh kg}^{-1}/772 \text{ Wh L}^{-1}$ (**Fig. 5d** and **Supplementary Table 5**). These results were added in the revised manuscript, demonstrating the practical viability of the high-mass-loading BNQS cathodes over the previous works on high-mass-loading cathodes. The reviewer’s constructive comment is highly appreciated, again.

[Revised manuscript]

Supplementary Fig. 19 Cross-sectional SEM image of the high-mass-loading BNQS electrode (thickness of 315 μm) with an areal-mass-loading of 60 mg cm^{-2} .

Supplementary Fig. 20 **a** GITT profiles of the BNQS cathode with an areal-mass-loading of $60\ mg\ cm^{-2}$ upon repeated current stimuli at a charge/discharge current density of $0.1\ C/0.1\ C$ ($= 1.16\ mA\ cm^{-2}$). **b** Calculation of Li ion diffusion coefficients of the cathode. **c** Discharge capacities under varied discharge current densities ($0.05\ C$ ($= 0.58\ mA\ cm^{-2}$) – $0.5\ C$ ($= 5.80\ mA\ cm^{-2}$)) at a fixed charge rate of $0.05\ C$.

Supplementary Fig. 23 Charge/discharge profiles of the pouch-type cell ($26 \times 24\ mm^2$ in size) composed of BNQS cathode (areal capacity of $12.1\ mAh\ cm^{-2}$)|separator|Li metal anode (areal capacity of $20\ mAh\ cm^{-2}$) at 1st, 30th, 50th, and 70th cycle. To enable development of the practical cell, the N/P ratio was 1.6, while the E/C ratio was set as $2.3\ g\ Ah^{-1}$.

Fig. 5 Enabling high-energy-density Li metal cells by the BNQS cathodes. a Charge/discharge profiles of the cells (BNQS cathode|separator|Li metal anode ($100\ \mu\text{m}$ corresponding to an areal capacity of $20\ \text{mAh cm}^{-2}$)) as a function of areal-mass-loading of the BNQS cathodes at a voltage range of $3.0 - 4.4\ \text{V}$ and charge/discharge current density of $0.05\ \text{C}/0.1\ \text{C}$. **b** Realization of theoretical specific capacities of cathode materials as a function of areal capacity (BNQS cathodes versus previously reported cathodes). **c** Gravimetric energy densities of cells as a function of areal capacity (BNQS cathodes versus previously reported cathodes), in which the cell weight was estimated by considering those of cathodes, anodes, separators, and electrolytes. **d** Cycling performance of the pouch-type cell ($26 \times 24\ \text{mm}^2$ in size). The cell consisted of BNQS cathode (areal capacity of $12.1\ \text{mAh cm}^{-2}$)|separator|Li metal anode (areal capacity of $20\ \text{mAh cm}^{-2}$), in which the N/P ratio was 1.6 and the amount of electrolyte in the cell (including the gel electrolyte in the BNQS cathode) was set as $2.3\ \text{g Ah}^{-1}$. The energy density (with packaging) of the pouch-type cell was estimated by considering its entire cell weight. The cell was reassembled by replacing the cycled Li metal anode with a fresh one after 70 cycles.

“The BNQS cathodes showed normal and stable charge/discharge behavior a voltage range of 3.0 – 4.2 V over the entire range of areal-mass-loadings explored herein (Supplementary Fig. 18), in which the BNQS cathodes were coupled with the Li metal anode (100 μm). The BNQS cathode with an areal-mass-loading of 60 mg cm^{-2} , of which cross-sectional morphology was shown in Supplementary Fig. 19, reached a (cathode sheet-based) areal capacity of 11.6 mAh cm^{-2} . The D_{Li^+} of the BNQS cathode was measured to be $1.63 \times 10^{-8} \text{ cm}^2 \text{ s}^{-1}$ (Supplementary Fig. 20a), which is comparable to that of the BNQS cathode with the areal-mass-loading of 36 mg cm^{-2} ($D_{\text{Li}^+} = 1.57 \times 10^{-8} \text{ cm}^2 \text{ s}^{-1}$, Fig. 3d). The discharge rate capability of the BNQS cathode with the areal-mass-loading of 60 mg cm^{-2} was provided in Supplementary Fig. 20b. The discharge capacity at a fast current density of 0.5 C was relatively low, which could be due to the high current density (0.5 C = 5.8 mA cm^{-2}) and long tortuous pathways of ions and electrons. Future studies will be devoted to further enhancing the rate capability via fine-tuning of electrode structure and materials chemistry. To increase the cell energy densities further, the charge cut-off voltage was raised to 4.4 V. The higher charge voltage led to an increase in the areal capacities, eventually reaching an areal capacity of 12.3 mAh cm^{-2} (Fig. 5a and Supplementary Fig. 21) which leads to a N/P ratio of 1.6 in the cell.”

“The aforementioned efficiency of capacity utilization in the electrodes crucially affects the energy densities of cells. The energy densities (without packaging) of the BNQS cathode-containing cells (404 $\text{Wh kg}^{-1}/1025 \text{ Wh L}^{-1}$) were compared with those of previously reported high-mass-loading cathode systems (Fig. 5c, Supplementary Fig. 22 and Supplementary Table 3), in which the energy densities were estimated without including packaging substances (see Supplementary Table 4 for calculation details). The highest energy densities (without packaging) achieved by the BNQS cathode with areal capacity of 12.3 mAh cm^{-2} exceeded those of previous studies, demonstrating the viability of the BNQS cathode approach in enabling the high-energy-density cell.

Furthermore, a pouch-type cell (26 \times 24 mm^2 in size) with the BNQS cathode was fabricated and its cell performance was investigated at a voltage range of 3.0 – 4.4 V to explore the feasibility of this high-mass-loading electrode concept. The pouch-type cell was composed of the BNQS cathode (areal capacity of 12.1 mAh cm^{-2})|separator|Li metal anode (areal

capacity of 20 mAh cm^{-2}), in which the N/P ratio was 1.6. The amount of electrolyte in the cell (including the gel electrolyte in the BNQS cathode) was limited as 2.3 g Ah^{-1} to practically evaluate the cell.^{44,54} The obtained pouch-type cell showed an initial discharge capacity of 203 mAh g^{-1} (Supplementary Fig. 23), exhibiting almost full utilization of the theoretical specific capacity (205 mAh g^{-1})^{30,31} of NCM811 in the BNQS electrode. In addition, the energy densities (with packaging) of the pouch-type cell were estimated to reach $321 \text{ Wh kg}^{-1}/772 \text{ Wh L}^{-1}$ (Fig. 5d and Supplementary Table 5). The pouch-type cell showed stable cycling performance (Fig. 5d and Supplementary Fig. 23), underscoring the electrochemical viability of the high-mass-loading BNQS cathodes. Meanwhile, no significant difference in the electrode thickness was observed before and after the cycling test (Supplementary Fig. 24), indicating the structural stability of the BNQS cathode with cycling. Intriguingly, the areal capacity of the pouch-type cell was almost recovered to an initial value, after the cycled Li metal anode was replaced with a fresh one after 70 cycles by referring to the procedure described in a previous work⁵⁵ (Fig. 5d). This result may indicate that the BNQS cathode is not a crucial cause of the cycling fading of the pouch-type cell.”

2. The electrode with 160 μm has a mass loading of 36 mg cm^{-2} . Why does that with $\sim 315 \mu\text{m}$ has the mass loading to 59.3 mg cm^{-2} ? The fabrication process is not linear?

→ Many thanks for the reviewer’s valuable comments. The deviation in the linearity between the electrode thickness and areal-mass-loading of electrode active materials could stem from the introduction of electroconductive-mat interlayers that were inserted between the electrode active layers and did not contribute to the areal-mass-loading. As described in the original manuscript, a unit BNQS electrode was composed of an electrode active layer and an electroconductive-mat interlayer. Thus, 8 electroconductive-mat interlayers were incorporated in a BNQS cathode with an areal-mass-loading of 60 mg cm^{-2} in comparison to that (4 layers) of a BNQS cathode with an areal-mass-loading of 36 mg cm^{-2} , thus causing the deviation in the linearity between the electrode thickness and areal-mass-loading. To clearly deliver the information on the BNQS cathode structure and components, the number, weight, and volume of electroconductive-mat interlayers (including the net weight/volume of the layers) were

provided as a function of areal-mass loadings (**Supplementary Table 1**).

[Revised manuscript]

Supplementary Table 1 Basic information of the electroconductive-mat interlayers inserted in the BNQS electrode as a function of areal-mass loading.

Areal-mass-loading of active materials (mg cm ⁻²)	Electroconductive-mat interlayers		
	Number	Areal-weight (mg cm ⁻²)	Thickness (μm)
16	1	0.8	9
23	2	1.2	18
36	3	1.6	27
51	6	2.8	54
60	7	3.2	63

“On top of the electroconductive mat that can act as an alternative porous current collector replacing the conventional metallic-foil one, the above-prepared electrode paste was stencil-printed, followed by UV curing, producing a BNQS unit electrode (areal-mass (solely including NCM811, if not specified) loading ~ 10 mg cm⁻²). The optimal thicknesses (40 ~ 60 μm) of active layers in the unit electrodes were determined by analyzing the electronic conductivity of the unit electrodes as a function of thickness of active layer (**Supplementary Fig. 4**). Subsequently, the stencil-printing-based electrode fabrication step was repeated, together with insertion of the electroconductive mats in the through-thickness direction, until reaching the mass-loading value of interest (**Fig. 1a**). After pressing to ensure intimate interfacial contact between the electrode components, a metallic-foil current-collector-free, electroconductive-mat/gel-electrolyte-embedded BNQS electrode was obtained (**Supplementary Fig. 5**). **The basic information of the electroconductive-mat interlayers inserted in the BNQS electrodes was provided as a function of areal-mass-loading (Supplementary Table 1).**”

3. The specific discharge capacity of 191 mAh/g is based on the mass of active material only

or the whole electrode? The gel electrolyte has a weight ratio of 20%.

→ Our deep appreciation is devoted to the reviewer's valuable comment. The specific discharge capacity of 191 mAh g⁻¹ was based on the mass of cathode active materials (here, NCM811) under the cut-off voltage of 4.2 V. In response to the reviewer's comment, a unit of specific discharge capacity was revised to avoid any unwanted misunderstanding, mAh g⁻¹ → mAh g_{NCM811}⁻¹. The reviewer's valuable comment is highly appreciated, again.

[Revised manuscript]

“The charge/discharge rate capabilities of the BNQS and slurry-cast cathodes with the same areal-mass-loading of 36 mg cm⁻² (Fig. 3a and Supplementary Fig. 11) were compared, in which the charge/discharge current rates varied from 0.05 (0.35 mA cm⁻²) to 0.5 C. Under the high areal-mass-loading, the BNQS cathode showed a specific discharge capacity of 191 mAh g_{NCM811}⁻¹ at 0.05 C, which almost reached the theoretical capacity (~192 mAh g_{NCM811}⁻¹)⁴⁶ of NCM811.”

As described in the original manuscript and also the reviewer pointed out, the composition ratio of the BNQS electrode already included the gel electrolyte, which was different from composition ratios (typically expressed as electrode active materials/conductive additives/binders) of conventional electrodes. Actually, to avoid this unwanted misunderstanding, this issue was mentioned in the original manuscript and two different composition ratios of the BNQS electrode were provided. Excluding the liquid electrolyte from the gel electrolyte led to an electrode composition of NCM811/carbon black/ETPTA (acting as a kind of binder) = 90.5/6/3.5 (w/w/w), revealing that the NCM811 content in the BNQS electrode appeared comparable to those of previously reported electrodes. Please see the sentences below.

[Original manuscript]

“We note that the BNQS electrode already includes 20 wt.% gel electrolyte (comprising the UV-cured ETPTA polymer and 1M LiPF₆ in EC/PC) besides NCM811 and carbon black additives, which could correspond to binders and liquid electrolytes embedded in pores of conventional LIB electrodes. The contents of binders and liquid electrolytes in conventional LIB electrodes are generally in the range of 15 – 20 wt.%,³⁵ which are not significantly different

from the gel electrolyte content of the BNQS electrode. Excluding the liquid electrolyte from the gel electrolyte leads to an electrode composition of NCM811/carbon black/ETPTA (acting as a kind of binder) = 90.5/6/3.5 (w/w/w), revealing that the NCM811 content in the BNQS electrode appears comparable to those of previously reported electrodes.”

4. In figure 2E, it is hard to directly attribute the better ion transport behavior of BNQS to the structural effect. The authors should provide more evidence about it. Moreover, why the BNQS electrode has a larger semi-circle? More explanations should be added.

→ Many thanks for the reviewer’s valuable comment. As the reviewer is aware of, the ionic resistance of electrodes can be measured using the symmetric cell configuration at a fully lithiated state (ref.: *Science*, **356**, 599, (2017) and *Electrochemistry*, **75**, 7, (2005)). As described in the original manuscript, Ogihara *et al.* (ref. [40, 41]) reported that the projection of a slope (observed in the low-frequency region of EIS profiles) to a real axis, which is defined as ionic resistance (R_{ion})/3, reflects the ionic resistance inside the electrodes. In **Fig. 2e**, the BNQS electrode showed a lower $R_{ion}/3$ ($6.43 \Omega \text{ cm}^2$) than the slurry-cast electrode ($14.81 \Omega \text{ cm}^2$). This lower ionic resistance of the BNQS electrode was ascribed to the structural effect of the BNQS electrode. The structural difference between the BNQS and slurry-cast electrodes was elucidated in **Figs. 2a-c**. In comparison to the slurry-cast electrode showing the nonuniform distribution of electrode components (resulting in the nonuniform distribution of porous structure that will be filled with electrolytes) along the through-thickness direction, the BNQS electrode exhibited the homogeneous distribution of electrode components, beneficially contributing to the formation of highly interconnected ion conduction channels in the through-thickness direction. In response to the reviewer’s comment on the semi-circle in the EIS spectra, we quantitatively analyzed the Nyquist plot using a transmission line equivalent circuit model (ref.: *Science*, **356**, 599, (2017) and *Adv. Funct. Mater.*, **29**, 1809196, 2019). The obtained semi-circle values (R_{high} , defined as ionic or electric resistance of the interface between electrolyte and electrode or between the electrode and current collector, ref.: *Science*, **356**, 599, (2017) and *Adv. Funct. Mater.*, **29**, 1809196, 2019) were $14.1 \Omega \text{ cm}^2$ for the BNQS electrode and $14.2 \Omega \text{ cm}^2$ for the slurry-cast electrode, exhibiting no significant difference in the R_{high} between the

two electrodes.

Fig. S7. Nyquist plots for composites with tunable in-plane nanopores by using a symmetric cell with two identical electrodes (11 mg cm^{-2}) for a state of charge (SOC) at 0% (hollow symbols) (A-D). The solid lines are the best-fitting simulations for the equivalent circuits using the generalized finite length Warburg element open circuit terminus (W_o) as shown in (E) (36). W_o is used to simulate $R_{ion}/3$, and its fitting parameter R_{W_o} corresponds to the value of $R_{ion}/3$. The different impedances determined as $R_{ion}/3$, R_{high} and R_{sol} for the various electrodes are listed in the table S1. The inset in (B) shows how $R_{ion}/3$ is determined.

Fig. S8. Transmission line equivalent circuit of porous electrodes. (A) Non-faradaic process at un lithiated state (SOC = 0%) for porous electrodes. (B) Faradaic process at lithiated state (SOC > 0%). R_{sol} is the ohmic resistance originating primarily from the electrolyte; R_{high} is ionic or electric resistance of the interface between the electrolyte and electrode or between the electrode and current collector; R_{ion} is the ionic resistance in the electrolyte-filled porous electrode; C_{dl} and R_{ct} are the double layer capacitance and charge-transfer resistance, respectively.

Figure. A captured image from a research article published in *Science*, **356**, 599, (2017).

Figure 3. a) Nyquist plots for LFMP-IGF and LFMP-TGF composites by using a symmetric cell with two identical electrodes (8.5 mg cm^{-2}) for a state of charge (SOC) at 0%. b) Comparison of tortuosity, porosity, and calculated Li-ion and electron conductivities of LFMP-IGF and LFMP-TGF. The values of the parameters in the figure were calculated relative to LFMP-TGF. c) Illustration of LFMP-IGF (left) with vertical channels for short Li-ion transfer path and good electrolyte diffusion and (right) disorder structure of LFMP-TGF with poor electrolyte diffusion and longer Li-ion transfer path.

Calculation of Tortuosity

To fit the experimental data, we followed the previously reported method^[2], using the finite length Warburg element open circuit terminus (W_o) for the TLM and the equivalent circuits being shown as:

According to ref 2, the interfacial charge transfer resistance of graphene in LFMP-IGF/TGF, which accounts for the depressed semicircle in the high frequency region, is denoted as R_{high} . R_{sol} represents the electrolyte resistance and W_o is used to simulate $R_{ion}/3$. Hence, we obtain R_{ion} from this fitting, which is 14.1 ohm cm^2 for LFMP-TGF and 6.3 ohm cm^2 for LFMP-IGF.

After substituting the values of R_{ion} , κ , L and ε into the calculation equation, the tortuosity of LFMP-IGF and LFMP-TGF are 2.5 and 4.1, respectively. It's worth noting that the tortuosity calculated from EIS measurements is higher than those derived from 3D reconstructions based on X-ray tomography, which would be attributed to the insufficient resolution of 3D reconstructions for the unresolved binder and conductive carbon phases^[3].

Therefore, the electrode tortuosity could be underestimated in that case.

Figure. Captured images from a research article published in *Adv. Funct. Mater.*, **29**, 1809196, (2019).

[Revised manuscript]

“In addition to the electron conduction networks mentioned above, ion conduction channels play an important role in achieving uniform redox reactions in high-mass-loading electrodes. To elucidate the ion transport phenomena inside the BNQS electrode, electrochemical impedance spectroscopy (EIS) analysis was conducted using a symmetric cell (electrode|separator|electrode) at a fully lithiated state (Fig. 2e). The Nyquist plot and corresponding simulation curves fitted by a transmission line equivalent circuit model (TLM)^{18,39} were provided in Supplementary Fig. 7. Ogihara *et al.*^{40,41} reported that the projection of a slope (observed in the low-frequency region of EIS profiles) to a real axis, which is defined as ionic resistance (R_{ion})/3, reflects the ionic resistance inside the electrodes. The BNQS electrode showed a lower $R_{ion}/3$ (6.43 Ω cm²) than the slurry-cast electrode (14.81 Ω cm²). Meanwhile, the gel electrolyte inside the BNQS electrode provided slightly lower ionic conductivity (2.73 mS cm⁻¹) than the liquid electrolyte (4.90 mS cm⁻¹) inside the slurry-cast electrode (Supplementary Fig. 8). Such inconsistency between the ionic resistance inside the electrodes and the bulk ionic conductivity of the electrolytes reveals that the ion transport phenomena inside electrodes could be more influenced by their structural effect (*i.e.*, homogeneity of ion conduction channels). As shown in Fig. 2a-c, the BNQS electrode provided the uniform distribution of components along the through-thickness direction compared to the slurry-cast electrode. For the slurry-cast electrode, a relatively dense structure was formed in the top layer due to the severe aggregation of carbon conductive additives, thus hindering the access of Li⁺ into the electrode.⁴² In contrast, the BNQS electrode exhibited the homogeneous distribution of its components, beneficially contributing to the formation of highly interconnected ion conduction channels in the through-thickness direction.”

Supplementary Fig. 7 Nyquist plots of **a** slurry-cast and **b** BNQS electrodes obtained by a symmetric cell configuration with two identical electrodes at a state of charge (SOC) of 0 %, in which hollow symbols and solid lines represent experimental data and simulation results, respectively. **c** Transmission line equivalent circuit model (TLM).^{18,39}

5. The increase of the cut-off voltage of course will increase the capacity and energy density. However, it will also influence cycling performance. It should be consistent. It is not suitable to use 4.2 V to get long cycling life but use 4.4 V to obtain the high energy density. Meanwhile, only 80 cycles are very limited.

→ Many thanks for the reviewer's valuable comment. In response to the reviewer's comment, the cycling performance of the BNQS cathode with the areal-mass-loading of 60 mg cm⁻² was investigated using a pouch-type cell at a voltage range of 3.0 – 4.4 V. Please see **Fig. 5d** in the revised manuscript. The pouch-type cell was composed of the BNQS cathode (areal capacity of 12.1 mAh cm⁻²)|separator|Li metal anode (areal capacity of 20 mAh cm⁻²), in which the N/P

ratio was 1.6. The amount of electrolyte in the cell (including the gel electrolyte in the BNQS cathode) was limited as 2.3 g Ah^{-1} to practically evaluate the cell (ref.: *Nat. Energy*, **4**, 180, (2019) and *Joule*, **3**, 1094, (2019)). The resulting pouch-type cell showed a discharge capacity of 203 mAh g^{-1} , exhibiting almost full utilization of the theoretical specific capacity (205 mAh g^{-1} , ref. [30, 31]) of NCM811 in the BNQS electrode. In addition, the pouch-type cell showed stable cycling performance (**Figs. 5d** and **Supplementary Fig. 23**), underscoring the electrochemical viability of the high-mass-loading BNQS cathodes. Meanwhile, the areal capacity of the pouch-type cell was almost recovered to an initial value, after the Li metal anode was replaced with a fresh one by referring to the procedure described in a previous work (ref.: *Joule* **3**, 1662 (2019)) (**Fig. 5d**). This result may indicate that the BNQS cathode is not a crucial cause of the cycling fading of the pouch-type cell.

[Revised manuscript]

Supplementary Fig. 23 Charge/discharge profiles of the pouch-type cell ($26 \times 24 \text{ mm}^2$ in size) composed of BNQS cathode (areal capacity of 12.1 mAh cm^{-2})|separator|Li metal anode (areal capacity of 20 mAh cm^{-2}) at 1st, 30th, 50th, and 70th cycle. To enable development of the practical cell, the N/P ratio was 1.6, while the E/C ratio was set as 2.3 g Ah^{-1} .

Fig. 5 Enabling high-energy-density Li metal cells by the BNQS cathodes. **a** Charge/discharge profiles of the cells (BNQS cathode|separator|Li metal anode (100 μm corresponding to an areal capacity of 20 mAh cm^{-2})) as a function of areal-mass-loading of the BNQS cathodes at a voltage range of 3.0 – 4.4 V and charge/discharge current density of 0.05 C/0.1 C. **b** Realization of theoretical specific capacities of cathode materials as a function of areal capacity (BNQS cathodes versus previously reported cathodes). **c** Gravimetric energy densities of cells as a function of areal capacity (BNQS cathodes versus previously reported cathodes), in which the cell weight was estimated by considering those of cathodes, anodes, separators, and electrolytes. **d** Cycling performance of the pouch-type cell (26 \times 24 mm^2 in size). The cell consisted of BNQS cathode (areal capacity of 12.1 mAh cm^{-2})|separator|Li metal anode (areal capacity of 20 mAh cm^{-2}), in which the N/P ratio was 1.6 and the amount of electrolyte in the cell (including the gel electrolyte in the BNQS cathode) was set as 2.3 g Ah^{-1} . The energy density (with packaging) of the pouch-type cell was estimated by considering its entire cell weight. The cell was reassembled by replacing the cycled Li metal anode with a fresh one after 70 cycles.

“The aforementioned efficiency of capacity utilization in the electrodes crucially affects the energy densities of cells. The energy densities (without packaging) of the BNQS cathode-containing cells ($404 \text{ Wh kg}^{-1}/1025 \text{ Wh L}^{-1}$) were compared with those of previously reported high-mass-loading cathode systems (Fig. 5c, Supplementary Fig. 22 and Supplementary Table 3), in which the energy densities were estimated without including packaging substances (see Supplementary Table 4 for calculation details). The highest energy densities (without packaging) achieved by the BNQS cathode with areal capacity of 12.3 mAh cm^{-2} exceeded those of previous studies, demonstrating the viability of the BNQS cathode approach in enabling the high-energy-density cell.

Furthermore, a pouch-type cell ($26 \times 24 \text{ mm}^2$ in size) with the BNQS cathode was fabricated and its cell performance was investigated at a voltage range of $3.0 - 4.4 \text{ V}$ to explore the feasibility of this high-mass-loading electrode concept. The pouch-type cell was composed of the BNQS cathode (areal capacity of 12.1 mAh cm^{-2})|separator|Li metal anode (areal capacity of 20 mAh cm^{-2}), in which the N/P ratio was 1.6. The amount of electrolyte in the cell (including the gel electrolyte in the BNQS cathode) was limited as 2.3 g Ah^{-1} to practically evaluate the cell.^{44,54} The obtained pouch-type cell showed an initial discharge capacity of 203 mAh g^{-1} (Supplementary Fig. 23), exhibiting almost full utilization of the theoretical specific capacity (205 mAh g^{-1})^{30,31} of NCM811 in the BNQS electrode. In addition, the energy densities (with packaging) of the pouch-type cell were estimated to reach $321 \text{ Wh kg}^{-1}/772 \text{ Wh L}^{-1}$ (Fig. 5d and Supplementary Table 5). The pouch-type cell showed stable cycling performance (Fig. 5d and Supplementary Fig. 23), underscoring the electrochemical viability of the high-mass-loading BNQS cathodes. Meanwhile, no significant difference in the electrode thickness was observed before and after the cycling test (Supplementary Fig. 24), indicating the structural stability of the BNQS cathode with cycling. Intriguingly, the areal capacity of the pouch-type cell was almost recovered to an initial value, after the cycled Li metal anode was replaced with a fresh one after 70 cycles by referring to the procedure described in a previous work⁵⁵ (Fig. 5d). This result may indicate that the BNQS cathode is not a crucial cause of the cycling fading of the pouch-type cell.”

6. It is recommended to characterize the structure and morphology of the cathode after cycling.
 → Many thanks for the reviewer's valuable comment. The post-mortem analysis of the cycled cathode was already described in the original manuscript. Please see **Fig. 4c-f**. In response to the reviewer's comment, the cross-sectional SEM image of the cycled BNQS cathode (at a voltage range of 3.0 – 4.4 V) was newly added in revised manuscript (**Supplementary Fig. 24**). No significant difference in the electrode thickness was observed before and after the cycling test, indicating the structural stability of the BNQS cathode with cycling.

[Original manuscript]

Fig. 4 Cycling performance of the BNQS cathode (versus slurry-cast cathode) and post-mortem analysis. **c** TOF-SIMS mapping images of the NiF_2^+ byproducts formed on the surface

of the cathodes. **d** XPS F 1s spectra, focusing on a characteristic peak assigned to NiF₂ (684.6 eV) byproducts, of the cathodes. **e** EDS line intensities for P element along the depth of NCM811 particles. **f** HR-TEM images with fast Fourier transform patterns of the NCM811 particles for (left) slurry-cast and (right) BNQS cathodes.

[Revised manuscript]

“The pouch-type cell showed stable cycling performance (Fig. 5d and Supplementary Fig. 23), underscoring the viability of the high-mass-loading BNQS cathodes. Meanwhile, no significant difference in the electrode thickness was observed before and after the cycling test (Supplementary Fig. 24), indicating the structural stability of the BNQS cathode with cycling.”

Supplementary Fig. 24 Cross-sectional SEM image of the BNQS cathode (areal-mass-loading of 60 mg cm⁻²) after the cycling test (shown in Fig. 5d).

7. Is Figure 5a based on coin cell or pouch-type cell? The data of 59.3 mg cm⁻² is almost the same with that in Figure 5d. It is not possible to get the same data from coin cell and pouch-type cell. The authors should compare them.

→ Our deep appreciation is devoted to the reviewer’s valuable comments. The cell configurations shown herein were a coin-type cell (Fig. 5a) and a pouch-type cell (Fig. 5d and

Supplementary Fig. 23), respectively. Please note that the pouch-type cell showed a slightly lower specific capacity ($203 \text{ mAh g}_{\text{NCM811}}^{-1}$) than that ($205 \text{ mAh g}_{\text{NCM811}}^{-1}$) of the coin cell. To address the reviewer's concern, the charge/discharge behavior of the pouch-type cell was investigated again. A similar value of the specific capacity was observed. Please see the results below.

Figure. a Charge/discharge profiles of the coin and pouch-type cell containing BNQS cathode (areal-mass-loading of 60 mg cm^{-2}) at a voltage range of 3.0 – 4.4 V and charge/discharge current density of 0.05 C/0.1 C. **b** High-magnification view of the discharge profiles of the cells.

[Revised manuscript]

Supplementary Fig. 23 Charge/discharge profiles of the pouch-type cell ($26 \times 24 \text{ mm}^2$ in size)

composed of BNQS cathode (areal capacity of 12.1 mAh cm^{-2})|separator|Li metal anode (areal capacity of 20 mAh cm^{-2}) at 1st, 30th, 50th, and 70th cycle. To enable development of the practical cell, the N/P ratio was 1.6, while the E/C ratio was set as 2.3 g Ah^{-1} .

8. The cycling performance of pouch-type cells should be characterized to show the potential of the practical applications.

→ Many thanks for the reviewer's valuable comments. In response to the reviewer's comment, the cycling performance of the BNQS cathode with the areal-mass-loading of 60 mg cm^{-2} was examined using a pouch-type cell at a voltage range of 3.0 – 4.4 V. Please see the newly revised **Fig. 5d**. The pouch-type cell was composed of the BNQS cathode (areal capacity of 12.1 mAh cm^{-2}) |separator|Li metal anode (areal capacity of 20 mAh cm^{-2}), in which the N/P ratio was 1.6. . The amount of electrolyte in the cell (including the gel electrolyte in the BNQS cathode) was limited as 2.3 g Ah^{-1} to practically evaluate the cell (ref.: *Nat. Energy*, **4**, 180, (2019), *Joule*, **3**, 1094, (2019)). The resulting pouch-type cell showed a discharge capacity of 203 mAh g^{-1} , exhibiting almost full utilization of the theoretical specific capacity (205 mAh g^{-1} , ref. [30, 31]) of NCM811 in the BNQS electrode. In addition, the pouch-type cell showed stable cycling performance (**Fig. 5d** and **Supplementary Fig. 23**), underscoring the electrochemical viability of the high-mass-loading BNQS cathodes. Moreover, the energy densities (with packaging) of the pouch-type cell were estimated to reach $321 \text{ Wh kg}^{-1}/772 \text{ Wh L}^{-1}$ (**Fig. 5d** and **Supplementary Table 5**).

[Revised manuscript]

Fig. 5 Enabling high-energy-density Li metal cells by the BNQS cathodes. **a** Charge/discharge profiles of the cells (BNQS cathode|separator|Li metal anode (100 μm corresponding to an areal capacity of 20 mAh cm^{-2})) as a function of areal-mass-loading of the BNQS cathodes at a voltage range of 3.0 – 4.4 V and charge/discharge current density of 0.05 C/0.1 C. **b** Realization of theoretical specific capacities of cathode materials as a function of areal capacity (BNQS cathodes versus previously reported cathodes). **c** Gravimetric energy densities of cells as a function of areal capacity (BNQS cathodes versus previously reported cathodes), in which the cell weight was estimated by considering those of cathodes, anodes, separators, and electrolytes. **d** Cycling performance of the pouch-type cell (26 \times 24 mm^2 in size). The cell consisted of BNQS cathode (areal capacity of 12.1 mAh cm^{-2})|separator|Li metal anode (areal capacity of 20 mAh cm^{-2}), in which the N/P ratio was 1.6 and the amount of electrolyte in the cell (including the gel electrolyte in the BNQS cathode) was set as 2.3 g Ah^{-1} . The energy density (with packaging) of the pouch-type cell was estimated by considering its entire cell weight. The cell was reassembled by replacing the cycled Li metal anode with a fresh one after 70 cycles.

Supplementary Fig. 23 Charge/discharge profiles of the pouch-type cell ($26 \times 24 \text{ mm}^2$ in size) composed of BNQS cathode (areal capacity of 12.1 mAh cm^{-2})|separator|Li metal anode (areal capacity of 20 mAh cm^{-2}) at 1st, 30th, 50th, and 70th cycle. To enable development of the practical cell, the N/P ratio was 1.6, while the E/C ratio was set as 2.3 g Ah^{-1} .

“The aforementioned efficiency of capacity utilization in the electrodes crucially affects the energy densities of cells. The **energy densities (without packaging)** of the BNQS cathode-containing cells ($404 \text{ Wh kg}^{-1}/1025 \text{ Wh L}^{-1}$) were compared with those of previously reported high-mass-loading cathode systems (**Fig. 5c, Supplementary Fig. 22 and Supplementary Table 3**), in which the **energy densities were estimated without including packaging substances** (see **Supplementary Table 4** for calculation details). **The highest energy densities (without packaging) achieved by the BNQS cathode with areal capacity of 12.3 mAh cm^{-2}** exceeded those of previous studies, demonstrating the viability of the BNQS cathode approach in enabling the high-energy-density cell.

Furthermore, a pouch-type cell ($26 \times 24 \text{ mm}^2$ in size) with the BNQS cathode was fabricated **and its cell performance was investigated at a voltage range of $3.0 - 4.4 \text{ V}$** to explore the feasibility of this high-mass-loading electrode concept. **The pouch-type cell was composed of the BNQS cathode (areal capacity of 12.1 mAh cm^{-2})|separator|Li metal anode (areal capacity of 20 mAh cm^{-2})**, in which the N/P ratio was 1.6. The amount of electrolyte in the cell

(including the gel electrolyte in the BNQS cathode) was limited as 2.3 g Ah^{-1} to practically evaluate the cell.^{44,54} The obtained pouch-type cell showed an initial discharge capacity of 203 mAh g^{-1} (Supplementary Fig. 23), exhibiting almost full utilization of the theoretical specific capacity (205 mAh g^{-1})^{30,31} of NCM811 in the BNQS electrode. In addition, the energy densities (with packaging) of the pouch-type cell were estimated to reach $321 \text{ Wh kg}^{-1}/772 \text{ Wh L}^{-1}$ (Fig. 5d and Supplementary Table 5). The pouch-type cell showed stable cycling performance (Fig. 5d and Supplementary Fig. 23), underscoring the electrochemical viability of the high-mass-loading BNQS cathodes. Meanwhile, no significant difference in the electrode thickness was observed before and after the cycling test (Supplementary Fig. 24), indicating the structural stability of the BNQS cathode with cycling. Intriguingly, the areal capacity of the pouch-type cell was almost recovered to an initial value, after the cycled Li metal anode was replaced with a fresh one after 70 cycles by referring to the procedure described in a previous work⁵⁵ (Fig. 5d). This result may indicate that the BNQS cathode is not a crucial cause of the cycling fading of the pouch-type cell.”

REVIEWER COMMENTS

Reviewer #1 (Remarks to the Author):

Kim and co-workers dedicated great efforts to addressing the comments from all referees. These revisions not only strengthen the manuscript substantially, but also improve the readability of the manuscript. The referee was impressed by and satisfied with the revisions the authors made and would like to recommend acceptance of this interesting and impactful work for publication in the journal of Nature Communications.

By Chaoji Chen, Wuhan University

Reviewer #2 (Remarks to the Author):

Although authors indicated some reported papers also calculated energy densities without the weight of package, the cell's energy densities of 321wh/kg with package was shown in revised Supplementary Table 5 which was a progress. However, the 80 cycles shown in figure 5d indicated poor performance comparing with the 1000 cycles of all-solid-state Li rechargeable batteries which were reported in Nature Energy (10.1038@s41560-020-0575-z). Moreover, the thickness of the cell should be changed with cycles, so the volumetric energy densities 772Wh/L (at same level of now's Li-ion batteries with liquid electrolyte), should be indicated at which states (discharge/or charge state?). According the electrochemical performance in the manuscript, this BNQS cell's properties can't satisfy the high level of Nature Communications. It is not suitable for publication at now's status.

Reviewer #3 (Remarks to the Author):

The authors devoted many efforts to improving the quality of the work. The reviewer is pleased with the revisions and responses. Hence, it is recommended to be published in Nature Communications.

Reviewer's Comments

Reviewer #1

Kim and co-workers dedicated great efforts to addressing the comments from all referees. These revisions not only strengthen the manuscript substantially, but also improve the readability of the manuscript. The referee was impressed by and satisfied with the revisions the authors made and would like to recommend acceptance of this interesting and impactful work for publication in the journal of Nature Communications.

→ Our deep appreciation is devoted to the reviewer's insightful comment.

Reviewer #3

The authors devoted many efforts to improving the quality of the work. The reviewer is pleased with the revisions and responses. Hence, it is recommended to be published in Nature Communications.

→ Our deep appreciation is devoted to the reviewer's constructive comment.

Reviewer #2

Although authors indicated some reported papers also calculated energy densities without the weight of package, the cell's energy densities of 321 wh/kg with package was shown in revised Supplementary Table 5 which was a progress.

→ Our deep appreciation is devoted to the reviewer's valuable comment.

1. However, the 80 cycles shown in figure 5d indicated poor performance comparing with the 1000 cycles of all-solid-state Li rechargeable batteries which were reported in Nature Energy ([10.1038@s41560-020-0575-z](https://doi.org/10.1038@s41560-020-0575-z)).

→ Many thanks for the reviewer's valuable comment. Please note that the paper (*Nat. Energy* **5**, 299 (2020)) mentioned by the reviewer showed the cycling performance using the low areal-capacity cathode (6.8 mAh cm⁻²), in sharp contrast to our study focusing on the high-areal-capacity electrode (12.1 mAh cm⁻²). In addition to this difference in the areal capacity, the paper of *Nat. Energy* showed the cycling performance at a high temperature of 60°C, in comparison to our results measured at 25°C. These conditions (i.e., low areal-capacity cathode and high measurement temperature) of the paper of *Nat. Energy* are favorable for enhancing cycling retention, as already reported in previous studies (ref.: *Joule*, **3**, 1094, (2019) and *Angew. Chem. Int. Ed.* **58**, 11364 (2019)). More notably, the paper of *Nat. Energy* did not provide any cyclability data at 25°C. Therefore, simple comparison of cycle numbers between this study and the paper of *Nat. Energy* without considering the areal-capacity of cathodes and measurement condition is not fair, I'm afraid.

Below is the summary underlying the difference between this study and the previous paper of *Nat. Energy* mentioned by the reviewer.

	This study	Nat. Energy 5 , 299 (2020)
Areal capacity of cathodes (mAh cm ⁻²)	12.1	6.8
Measurement temperature (°C)	25	60
Cycling performance at 25°C	Provided	None

Fig. 6 | Electrochemical performance of ASSB with SSEs. **a**, Characterization of a 0.6 Ah class prototype pouch cell and illustration of a bi-cell structure. **b**, X-ray CT of the bi-cell and symmetric structure based on an aluminium current collector. **c**, Schematic of pressurization process during the fabrication and operation of ASSB. After the cell assembly and stacking, pressurization was applied by using the WIP. During operation, an external pressure of 2 MPa was uniformly applied to the prototype pouch cell using a pressure jig. **d,e**, Rate capability of ASSBs at 60 °C. A 0.1C constant current and 4.25 V constant voltage (CC-CV) mode was adopted for all the charging sequences. Voltage profiles were plotted with cell capacity (**d**) and specific capacity (**e**). **f**, Discharge capacities were monitored under 0.1C/0.1C charge/discharge conditions as the discharging temperature was varied from 60 to -10 °C. The charging temperature was fixed at 60 °C. **g**, Cycling performance and Coulombic efficiency of the Ag-C|SSE|NMC prototype pouch cell (0.6 Ah) are plotted against the cycle numbers. A CC mode with the charge/discharge rate of 0.5C/0.5C was applied (voltage window, 2.5-4.25 V versus Li⁺/Li at 60 °C). The areal capacity loading of the NMC cathode was 6.8 mAh cm⁻² (1.0 C = 6.8 mA cm⁻²).

Figure. A captured image from a research article published in *Nat. Energy* 5, 299 (2020), “High-energy long-cycling all-solid-state lithium metal batteries enabled by silver–carbon composite anodes”.

In the 2nd revised manuscript, we added the longer cycling retention compared to the result of the 1st revised manuscript.

Fig. 5 Enabling high-energy-density Li metal cells by the BNQS cathodes. **d** Cycling performance of the pouch-type cell (26 × 24 mm² in size). The cell consisted of BNQS cathode (areal capacity of 12.1 mAh cm⁻²)|separator|Li metal anode (areal capacity of 20 mAh cm⁻²), in which the N/P ratio was 1.6 and the amount of electrolyte in the cell (including the gel

electrolyte in the BNQS cathode) was set as 2.3 g Ah⁻¹. The **specific energy** (with packaging) of the pouch-type cell was estimated by considering its entire cell weight. The cell was reassembled by replacing the cycled Li metal anode with a fresh one after 70 cycles.

2. Moreover, the thickness of the cell should be changed with cycles, so the volumetric energy densities 772Wh/L (at same level of now's Li-ion batteries with liquid electrolyte), should be indicated at which states (discharge/or charge state?). According the electrochemical performance in the manuscript, this BNQS cell's properties can't satisfy the high level of Nature Communications. It is not suitable for publication at now's status.

→ In response to the reviewer's comment, the cell state (here, discharge state) used to measure the volumetric energy density of the pouch-type cell was newly added. The reviewer's valuable comment is highly appreciated, again.

[Revised Manuscript]

“In addition, the **specific energy/energy density** (with packaging) of the pouch-type cell were estimated to reach 321 Wh kg⁻¹/772 Wh L⁻¹ **at the discharge state** (**Fig. 5d** and **Supplementary Table 5**). The pouch-type cell showed stable cycling performance (**Fig. 5d** and **Supplementary Fig. 23**), underscoring the electrochemical viability of the high-mass-loading BNQS cathodes.”

Supplementary Table 5. Parameters of the pouch-type cell, in which the cell was composed of the BNQS cathode (BNQS cathode (areal capacity of 12.1 mAh cm⁻²)|separator|Li metal anode (areal capacity of 20 mAh cm⁻²). The area of the BNQS cathode, Li metal anode and separator were 20 × 20 mm², 21 × 21 mm² and 23 × 23 mm², respectively. The **specific energies and energy densities** were estimated based on the experimentally measured weight and volume of the cell (including packaging substances).

Cell components	Mass (mg)	Areal mass (mg cm ⁻²)	Areal capacity (mAh cm ⁻²)	Thickness (μm)
BNQS cathode	330	83	12	314

Li-metal anode	64	13	20	109
Separator	7	1	-	16
Injected electrolyte	59	-	-	-
Pouch substances	106	-	-	150

Pouch-type cell parameters	Discharge state (Voltage = 3.0 V)
Mass (mg)	566
Thickness (μm)	589
Capacity (mAh)	48
Energy (mWh)	182
Specific energy (Wh kg^{-1})	321
Energy density (Wh L^{-1})	772

In regard to the reviewer's comment on the volumetric energy density of cells, current commercial 18650 cylindrical cells with liquid electrolytes are known to show energy densities of $\sim 650 \text{ Wh L}^{-1}$ (ref: *Nat. Commun.* **11**, 2499 (2021), *Nat. Commun.* **12**, 5459 (2021), and *Nat. Rev. Mater.* **1**, 16013, (2016)). As the reviewer is aware of, energy densities of 18650 cylindrical cells are known to be approximately 20% higher than those of pouch-type cells (ref: *Nat. Rev. Mater.* **1**, 16013, (2016)). Please note that the energy density (772 Wh L^{-1}) of this study was obtained using a pouch-type cell. In addition, the cell energy density of our study was achieved with the high-areal-capacity cathode that is difficult to attain with previous works. The comparison in the specific energies between this study and the previously reported high-

mass-loading cathodes was shown in **Fig. 5c**, **Supplementary Fig. 22**, and **Supplementary Table 3**. Please see the revised manuscript.

Full-cell Li-ion batteries

Asahi Kasei Corporation assembled a full rechargeable battery combining the petroleum coke anode with Goodenough's LiCoO_2 cathode, which was later commercialized by Sony in 1990 ($\sim 80 \text{ Wh kg}^{-1}$, 200 Wh L^{-1}) (Fig. 1). The finding of Sanyo's researchers^{6,15} and Dahn's work¹⁶ with EC as co-solvent paved the way for the development of Li-ion batteries with a graphite anode and increased the voltage and energy density to 4.2 V and 400 Wh L^{-1} respectively. In 1993, Guyomard and Tarascon¹⁸ reported a new electrolyte formulation, LiPF_6 in EC/DMC, for its improved oxidation stability (Fig. 1b). This electrolyte remains one of the popular electrolytes until today, affording LiCoO_2 -based Li-ion batteries three times higher energy density (250 Wh kg^{-1} , 600 Wh L^{-1}) than that of the first-generation devices by Sony³.

Figure. A captured image from a research article published in *Nat. Commun.* **11**, 2499 (2021), "A retrospective on lithium-ion batteries".

Fig. 1 Overview of the development of LIBs over the years. **A** Evolution of LIBs from the rocking-chair battery concept to today's LIBs and next-generation Si/Si-B/Si-D||IC batteries. Key indicators (specific energy, energy density and cycle life) are comparatively displayed. **B** Research trend in the field of Si/Si-B/Si-D||IC cells. (Last updated on June 28th, 2021 at 2 pm, Berlin time). The key search words used in Scopus® were 'lithium-ion battery + silicon anode/ Si-based/derivatives + insertion and/or intercalation cathode'.

Figure. A captured image from a research article published in *Nat. Commun.* **12**, 5459 (2021), "Production of high-energy Li-ion batteries comprising silicon-containing anodes and insertion-type cathodes." The energy density of current state of LIBs and those of prospective LIBs.

Commercial cell configurations

Before addressing each of the various post-LIBs, we first discuss the different structures of commercial cells. In large-scale applications (for example, in electric vehicles), a certain number of cells are packed into a module. The design of the modules depends largely on the size and shape of the products, as well as their interconnecting circuits, safety and temperature control aspects. We restrict the scope of this Review to the material properties and behaviour at the single-cell level.

Current commercial cells adopt three cell types: cylindrical, prismatic and pouch (FIG. 1). Cylindrical cells in most products (including those used for Tesla Motors' vehicles) follow a standard model in terms of size — namely, the 18650 cell. Typical 18650 cells in commercial LIB products hold volumetric energy densities of 600–650 Wh l⁻¹, which are ~20% higher than those of their prismatic and pouch counterparts^{10,11} because a stacked cell assembly in a cylindrical cell is wound with a higher tension. The energy density of battery systems

Figure. A captured image from a research article published in *Nat. Rev. Mater.* **1**, 16013, (2016), “Promise and reality of post-lithium-ion batteries with high energy densities”. The energy densities of 18650 cylindrical cells are known to be approximately 20% higher than those of the pouch-type cells.

REVIEWERS' COMMENTS

Reviewer #2 (Remarks to the Author):

My comments have been addressed. The cycle has be also improved. Now, this manuscript can be recommended for publication.